# BLOCK-RECURRENT DYNAMICS IN VITS

**Mozes Jacobs**[*]
Harvard University

**Thomas Fel**[*]
Harvard University

**Richard Hakim**[*]
Harvard University

**Alessandra Brondetta**
Osnabrück University

**Demba Ba**
Harvard University

**T. Andy Keller**
Harvard University

## ABSTRACT

As Vision Transformers (ViTs) become standard backbones across vision, a mechanistic account of their computational phenomenology is now essential. Despite architectural cues that hint at dynamical structure, no settled framework interprets Transformer depth as a well-characterized flow. In this work, we introduce the **Block-Recurrent Hypothesis (BRH)**, arguing that trained ViTs admit a block-recurrent depth structure such that the computation of the original $L$ blocks can be accurately rewritten using only $k \ll L$ distinct blocks applied recurrently. Across diverse ViTs, between-layer representational similarity matrices suggest few contiguous phases. To determine whether these phases reflect reusable computation, we operationalize our hypothesis in the form of block recurrent surrogates of pretrained ViTs, which we call Recurrent Approximations to Phase-structured TransfORmers (Raptor). Using small-scale ViTs, we demonstrate that phase-structure metrics correlate with our ability to accurately fit Raptor and identify the role of training and stochastic depth in promoting the recurrent block structure. We then provide an empirical existence proof for BRH in foundation models by showing that we can train a Raptor model to recover $96\%$ of DINOv2 ImageNet-1k linear probe accuracy in only 2 blocks while maintaining equivalent runtime. To provide a mechanistic account of these observations, we leverage our hypothesis to develop a program of **Dynamical Interpretability**. We find (*i*) directional convergence into class-dependent angular basins with self-correcting trajectories under small perturbations, (*ii*) token-specific dynamics, where cls executes sharp late reorientations while patch tokens exhibit strong late-stage coherence reminiscent of a mean-field effect and converge rapidly toward their mean direction, and (*iii*) a collapse of the update to low rank in late depth, consistent with convergence to low-dimensional attractors. Altogether, we find that a compact recurrent program emerges along the depth of ViTs, pointing to a low-complexity normative solution that enables these models to be studied through principled dynamical systems analysis.

## 1 INTRODUCTION

In the last decade, Transformers have become the default neural network architecture across machine learning communities, scaling favorably with data and compute (Vaswani et al., 2017; Kaplan et al., 2020). In particular, Vision Transformers (ViTs) (Dosovitskiy et al., 2020) have become the core architecture used in visual foundation modeling frameworks such as DINOv2 (Oquab et al., 2023; Darcet et al., 2023) and CLIP (Radford et al., 2021); and have come to dominate a wide range of visual tasks, from general visual feature extraction (He et al., 2021; Chiu et al., 2024; Yun, 2025), to diffusion (Peebles & Xie, 2023), image segmentation (Kirillov et al., 2023; Liu et al., 2024), and video processing (Arnab et al., 2021; Baldassarre et al., 2025). This increasing breadth of use motivates a move from empirical optimization to principled understanding.

Two pressures make this understanding urgent. First, safety-critical deployments (Wang & Chung, 2022; Alecu et al., 2022) demand mechanisms whose internal computation is inspectable (Losch et al., 2021), diagnosable (Adebayo et al., 2020), and verifiable (Tjeng & Tedrake, 2019) rather than

---

[*] Equal contribution.
Correspondence to {mozesjacobs,tfel,rhakim,takeller}@g.harvard.edu.
⦿ kempnerinstitute.github.io/raptor

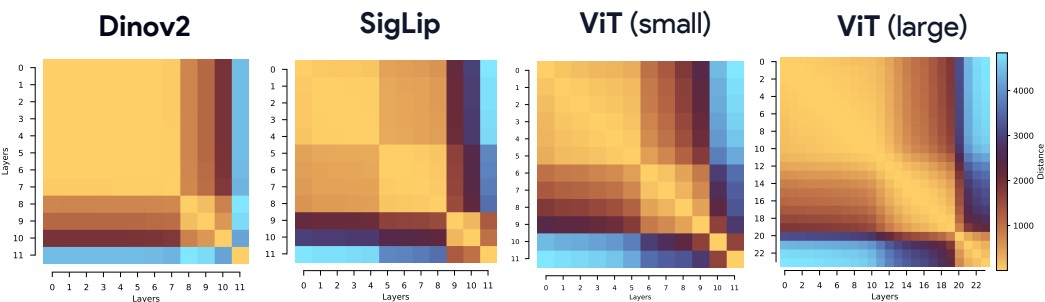

Figure 1: **Layer–layer similarity matrices across diverse Vision Transformers reveal block-structure.** Despite differences in scale and training objectives, all models exhibit contiguous block structure along depth, visible as phase-segmented regions of high similarity. Beyond representational similarity, this raises the question of whether a deeper *functional* recurrence underlies these patterns, hinting at block-wise reusability of computation across layers. In this work, we investigate this hypothesis, showing that these phase segments correspond to blocks with functional similarity, which can be approximated by a single shared block applied recurrently along depth.

opaque. As these models proliferate across domains, the ability to explain (Doshi-Velez & Kim, 2017; Gilpin et al., 2018; Kim et al., 2018), manipulate, and verify their behavior becomes increasingly essential. Second, from a scientific inference perspective (Cichy & Kaiser, 2019), understanding what makes these models work is essential for explaining their success. Their strong performance suggests they have discovered effective strategies, and identifying these strategies could advance our broader understanding of visual intelligence. Independently of any comparison to human vision, the goal is to uncover the principles that make these systems so effective.

One promising path toward such understanding is to search for underlying simplicity. Multiple approaches explore different facets of this simplicity, whether in functional expressivity (Montúfar et al., 2014; Telgarsky, 2015; Serra et al., 2018; Balestriero et al., 2018), symmetry (Cohen & Welling, 2014; Olah et al., 2020), or computation (Wilson, 2025; Goldblum et al., 2023; Schmidhuber, 1997; Mingard et al., 2025). Discovering such simplicity principle should improve both development (Bronstein et al., 2017; Frankle & Carbin, 2019) and interpretability (Bereska & Gavves, 2024; Carvalho et al., 2019; Fel, 2024; Ghorbani et al., 2017; Fel et al., 2023; Smilkov et al., 2017; Sundararajan et al., 2017; Zeiler & Fergus, 2014; Templeton et al., 2024; Bricken et al., 2023). Depth offers a concrete place to look for this simplicity. Residual connections have long suggested a link to dynamical systems (Liao & Poggio, 2020; Veit et al., 2016; Greff et al., 2016; Boulch, 2017; Haber & Ruthotto, 2017), hinting at implicit recurrence even when layers have distinct parameters. This convergent evidence makes plausible a form of algorithmic parsimony (Ma et al., 2022) in which a small set of blocks may be reused across many layers, trading parameters for iterations (Schwarzschild et al., 2021). Related perspectives support this view (Dingle et al., 2020). More concretely, residual updates invite a discrete-time interpretation of depth (Chen et al., 2018; Chalvidal et al., 2020; Sander et al., 2022), attention induces coupled token dynamics (Lu et al., 2019; Geshkovski et al., 2023), and in language models contiguous block recurrence has been observed and exploited (Geiping et al., 2025; Fernando & Guitchounts, 2025; Dehghani et al., 2018; Tan et al., 2023). This renewed interest in recurrent computation (Jolicoeur-Martineau, 2025; Venkatraman et al., 2025) and algorithmic complexity (Shaw et al., 2025; Dingle et al., 2020) as frameworks for understanding neural network simplicity bias further motivates our investigation. However, no existing framework characterizes depth in ViTs as representational flow or determines whether apparent phases correspond to functional reuse. Furthermore, vision explainability research (Bach et al., 2015; Fong & Vedaldi, 2017; Novello et al., 2022; Muzellec et al., 2024; Petsiuk et al., 2018; Hedström et al., 2022; Fel et al., 2025; Gorton, 2024; Kowal et al., 2024; Bau et al., 2017; Vilas et al., 2023) has not leveraged dynamical systems analysis to model emergent network structure.

In this work, we consider recurrence as a candidate source of simplicity and advance the Block-Recurrent Hypothesis (BRH): after training, the depth of a ViT organizes into a few contiguous phases such that the computation of the original $L$ layers can be rewritten by reusing only $k \ll L$ distinct blocks applied recurrently. Our starting point is an empirical observation: layer–layer representational similarity matrices consistently exhibit block-diagonal structure across disparate models. However, representational similarity does not necessarily translate to functional equivalence; therefore, we ask: *Does this phase structure reflect genuinely reusable computation?*

**Our contributions.** Our study proceeds in three parts:

- **Empirical evidence for block-recurrent structure.** We demonstrate across diverse ViTs that layer-layer representational similarity matrices exhibit distinct contiguous phases of computation, formalized through the Block-Recurrent Hypothesis. We develop a max-cut algorithm to systematically identify phase boundaries and show that this block structure is (***i***) associated with compressibility by a recurrent architecture and (***ii***) strengthened by stochastic depth.

- **Constructive verification via recurrent surrogates.** We operationalize the BRH by training weight-tied block-recurrent approximations of pretrained ViTs, termed `Raptor`. Critically, our goal is not compression or efficiency optimization per se, but rather to demonstrate that functional reuse is genuinely possible. `Raptor` reconstructs the complete internal representation trajectory across all layers, not merely the final output, providing strong evidence for true computational equivalence rather than input-output mimicry. Specifically, we provide empirical evidence for the BRH on foundational vision models by training a `Raptor` that recovers 96% of DINOv2's ImageNet-1k linear-probe accuracy using only 2 recurrent blocks, and 98% with 3 blocks.

- **Dynamical systems analysis framework.** Leveraging our hypothesis, we propose a program of Dynamical Interpretability that treats ViT depth as the discrete-time unfolding of an underlying dynamical system, such that the evolution of representations across layers can be analyzed through its temporal dynamics. Our analysis reveals: (***i***) directional convergence into class-dependent angular basins with self-correcting trajectories under perturbations, (***ii***) token-specific dynamics where `cls` tokens execute sharp late reorientations while patch tokens exhibit strong coherence reminiscent of mean-field behavior, and (***iii***) collapse of layer-to-layer updates to low-rank subspaces consistent with convergence to low-dimensional attractors.

As a first step, we characterize emergent phases in representation space, motivating the BRH.

## 2 EMERGENT PHASE STRUCTURE & THE BLOCK-RECURRENT HYPOTHESIS

Our investigation starts with a simple experiment: we construct layer-layer similarity matrices by computing the cosine similarity of each token at layer $l$ with the same token at layer $m$. As Figure 1 shows, despite varying tasks, objectives, and scales, all models exhibit consistent block-wise organization: high mutual similarity within contiguous blocks and lower similarity across boundaries. This finding is not new and echoes early (and more recent) observations in residual networks (Kornblith et al., 2019; Raghu et al., 2021; Nguyen et al., 2022; Hoang et al., 2025), but raises a fundamental question: does representational similarity reflect deeper computational structure? In fact, representational similarity alone provides no guarantee of functional equivalence. Layers might produce similar representations through entirely different computational pathways, or conversely, functionally equivalent computations might yield representations that appear dissimilar due to linear transformations or noise. The critical question is whether these apparent phases reflect genuine functional recurrence – reusing the same computational operations across layers within each phase. We formalize this possibility through the Block-Recurrent Hypothesis:

**Definition 1** (Block-Recurrent Hypothesis (BRH)). *We consider $\boldsymbol{f}$ to be a trained Vision Transformer with nominal depth $L$ and intermediate maps $\boldsymbol{f}_\ell : \mathcal{X} \to \mathcal{A}_\ell$, $\ell \in \{1, \dots, L\}$. We say that $\boldsymbol{f}$ satisfies the $\varepsilon$-BRH if for any $\ell$, there exist $k \ll \ell$ blocks $\boldsymbol{B}_1, \dots, \boldsymbol{B}_k$ and integers $n_1, \dots, n_k$ with $\sum_{j=1}^{k} n_j = \ell$ such that:*

$$\mathbb{E}_{\boldsymbol{x} \sim \mathbb{P}} \big( \| \boldsymbol{f}_\ell(\boldsymbol{x}) - (\boldsymbol{B}_k^{(n_k)} \circ \cdots \circ \boldsymbol{B}_1^{(n_1)})(\boldsymbol{x})) \|_F \big) \leq \varepsilon,$$

*where $\boldsymbol{B}_j^{(n_j)}$ denotes $n_j$ repeated applications of the same parameter-tied block $\boldsymbol{B}_j$ and the entire approximation maintaining equivalent computational cost.*

Here, $\| \cdot \|_F$ denotes the Frobenius norm and $\mathbb{P}$ is a probability distribution over natural images. To put it simply, this hypothesis states that a ViT's $L$ layers can be replaced by $k \ll L$ recurrent blocks that reproduce the entire internal trajectory at equivalent computational cost. By requiring intermediate layer fidelity rather than just final output matching, we rule out trivial solutions where computation is concentrated in a single block. The constraint $k \ll L$ with parameter tying ensures genuine functional reuse rather than simple parameter copying.

To test this hypothesis, we first operationalize it by proposing a method to construct these approximations using recurrent architectures, developing a specialized training technique described below.

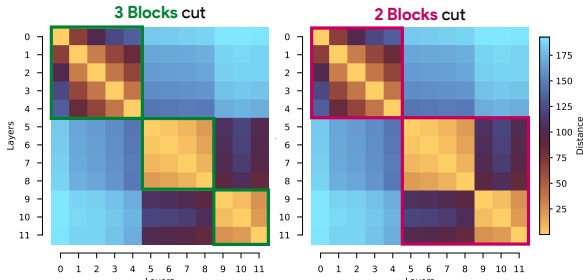 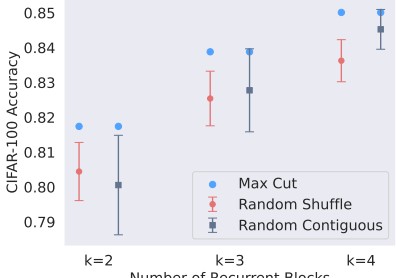

Figure 2: **Block discovery via max-cut segmentation of the layer–layer similarity matrix.** Our algorithm partitions depth into contiguous segments by maximizing within-block similarity and minimizing cross-block cosine similarity. Shown are two cuts of the same ViT-B: with 3-blocks (left, green) and 2-blocks (right, magenta). These cuts reveal candidate block boundaries where the representation dynamics undergo sharp transitions, providing an operational method for detecting contiguous recurrent phases in trained ViTs.

Figure 3: **Evaluation of `Raptor` models on CIFAR-100 using our max-cut partitioning algorithm versus random partitions.** Reported values are classification accuracy. Results for random partitions are aggregated over 10 different random partitions. Random shuffle refers to non-contiguous (fragmented) random partitions.

**Operationalizing Block-Recurrence with `Raptor`.** Since the BRH asserts only the existence of recurrent blocks satisfying its conditions and not their precise form, the most direct validation is constructive: demonstrating existence by example of a recurrent model approximating $\boldsymbol{f}$.

However, recurrent architectures are notoriously difficult to train: rollouts can drift as small state errors compound over steps, and gradients propagated through many recurrent applications often vanish or explode (Pascanu et al., 2013; Linsley et al., 2020; De et al., 2024). Standard backpropagation through time becomes unstable as recurrence depth increases, and the model must learn to simultaneously handle both the forward dynamics and its own prediction errors in closed loop (Williams & Zipser, 1989). To circumvent these challenges, we will leverage the intermediate layer activations as training targets, enabling a staged approach that combines the stability of teacher forcing with the self-consistency required for autoregressive deployment. More precisely, we introduce a procedure to distill existing Vision Transformers into Recurrent Approximations to Phase-structured TransfORmers (`Raptors`), using $k$ parameter-tied blocks with repetition counts determined by a max-cut phase discovery algorithm. This approach transforms the abstract hypothesis into a concrete architectural and training framework that can be empirically validated.

This constructive approach requires that `Raptor` models reproduce the internal activations of the full ViT they approximate, similar to Dasgupta & Cohn (2025); Sanh et al. (2019); Shleifer & Rush (2020), not merely mimic the final output[1]. The BRH implies that such reproduction should be possible within tolerance $\varepsilon$, making activation matching a natural training objective. Formally, let $\boldsymbol{f}$ be a reference ViT with intermediate activations $\boldsymbol{a}_\ell(\boldsymbol{x}) \equiv \boldsymbol{f}_\ell(\boldsymbol{x}) \in \mathbb{R}^{t \times d}$ for $\ell = 0, \dots, L$, where layer $\ell = 0$ denotes the patch encoder and $1 \le \ell \le L$ refer to transformer layers. Here, $t$ is the number of tokens and $d$ the feature dimension. Let $\boldsymbol{B}_j$ denote the $j$-th parameter-tied block in our recurrent decomposition. The `Raptor` approximation produces activations:

$$\tilde{\boldsymbol{a}}_\ell(\boldsymbol{x}) \equiv (\boldsymbol{B}_k^{(n_k)} \circ \cdots \circ \boldsymbol{B}_1^{(n_1)})(\boldsymbol{a}_0(\boldsymbol{x})) \tag{1}$$

where the composition covers layers 1 to $\ell$ according to our phase segmentation. We train `Raptor` using an autoregressive loss (AR) that enforces trajectory fidelity across all intermediate layers:

$$\mathcal{L}_h^{\text{AR}}(\boldsymbol{x}) = \mathbb{E}_{\boldsymbol{x}}\Big( \sum_{\ell=1}^{h} \|\tilde{\boldsymbol{a}}_\ell(\boldsymbol{x}) - \boldsymbol{a}_\ell(\boldsymbol{x})\|_F \Big), \quad h \le L. \tag{2}$$

This approach teaches each block to approximate its designated segment, while the overall model reproduces the teacher networks' complete representational trajectory. However, this formulation does not specify how to determine the block boundaries or phase assignments. We address this now with a simpler algorithmic approach based on the representational similarity structure observed earlier.

---

[1] Unlike classical distillation, which typically supervises logits (and occasionally a few intermediate "hints"), we enforce one-to-one alignment of all layers representations across the entire depth for the same inputs. The recurrent surrogate must generate the teacher's intermediate activations, not just its predictions.

**Choosing partitions.** For a given number of blocks $k$, we must introduce a practical method to determine the number of recurrent iterations of each block ($n_k$); in other words, where the recurrent "phases" of computation begin and end. We accomplish this by casting this "block discovery" process as a weighted max-cut problem Goemans & Williamson (1995), solved via dynamic programming (see Subsection A.1 for details). Specifically, the algorithm seeks to partition depth into contiguous segments by maximizing within-block similarity and minimizing cross-block similarity. We visualize the results of this procedure applied to ViT-B in Figure 2, demonstrating that the discovered blocks align reasonably with qualitative assessment.

To validate this approach, we train recurrent transformer models using max-cut partitions to reproduce the activations of trained Vision Transformers on CIFAR-100 (validation set accuracy 90.7%). Remarkably, as shown in Figure 3, Raptor student models with only 2 recurrent blocks closely match the performance of the ViT-B teacher models they approximate. The max-cut algorithm provides partitions that achieve strong performance, with accuracy often greater than one standard deviation above randomly chosen partitions. This observation suggests that the representational block-similarity structure is closely associated with functional block-recurrent phases.

To assess how unique each block-recurrent layer is and control for degenerate solutions to functional block-recurrence, we tested whether swapping a layer from a different block could substitute for a layer within a target block (Figure 15). Using DINOv2-B trained on ImageNet-1k, we find that while intra-block swapping preserves accuracy, inter-block swapping leads to model collapse. These results demonstrate that representational block-similarity structure predicts functional block-recurrent structure, and that layer identity is functionally unique to each block.

**How do blocks emerge?** Having operationalized the BRH and demonstrated a method for block discovery, we now turn to the mechanistic origins of this phenomenon: *under what conditions does this block-recurrent structure emerge in trained Vision Transformers*? To investigate this systematically, we examine small-scale ViTs where we control training conditions and isolate potential contributing factors. Specifically, we hypothesize that training and stochastic depth (Huang et al., 2016) may promote the emergence of block-recurrent patterns.

Motivated by evidence that residual networks tolerate variable effective depth (Wu et al., 2019), we examined the effect of stochastic depth (SD) on block recurrence. During training, each layer is dropped independently with probability $p$, applied uniformly across depth. We trained ViT-B/14 from random initialization on CIFAR-100, using the cls token for the linear probe across a sweep of SD $p$ rates. We observe an increase in layer-layer similarity with increasing SD $p$ rates (Figure 4A, E). Next, we used these trained ViT networks as teachers for student Raptor models (see Appendix A). Raptor models were trained to reconstruct the hidden activation states of the ViT teacher across layers. Raptor forward passes are fully autoregressive, meaning each layer's output is fed into the next layer and is also trained to match the corresponding layer in the teacher network (Figure 10). We quantify the similarity of the cls and patch token representations in each layer between the teacher and student networks as the $R^2$ of their matched token embeddings (Figure 4B).

We observe that, as ViT stochastic depth increases, a separately trained Raptor student model becomes significantly better at reconstructing the ViT's layerwise hidden states (Figure 4D). In addition, we observe that increasing SD improves CIFAR-100 classification accuracy in both ViT teacher and Raptor student networks (Figure 4C). We combine the above results in Figure 4E and observe a strong positive association between the ViT's layer-layer representational similarity and Raptor reconstruction fidelity. These results demonstrate that stochastic depth regularization increases layer-layer similarity and recurrent compressibility.

We also sought to determine whether recurrent compressibility was dependent on several other factors, including whether the ViT teacher was trained or untrained, and whether skip branches are included in the network architecture. We observe that network hidden layer activations in untrained ViTs can be reconstructed by Raptor better than trained ViTs. Interestingly, we observe that removing the skip branch from an untrained ViT only slightly reduces reconstruction accuracy. Finally, we trained Raptor students on checkpoints from a single ViT allowed to overfit (via reduced weight decay) on CIFAR-100 classification (Figure 13). We observe that Raptor reconstruction accuracy ($R^2$) remains high until the ViT network begins to overfit ( Figure 13B-D), after which cls token reconstruction and classification accuracy drops. These results add to the results on stochastic depth and demonstrate how the natural architecture of ViTs as well as the normative state of properly regularized ViT networks demonstrate emergent block recurrent dynamics.

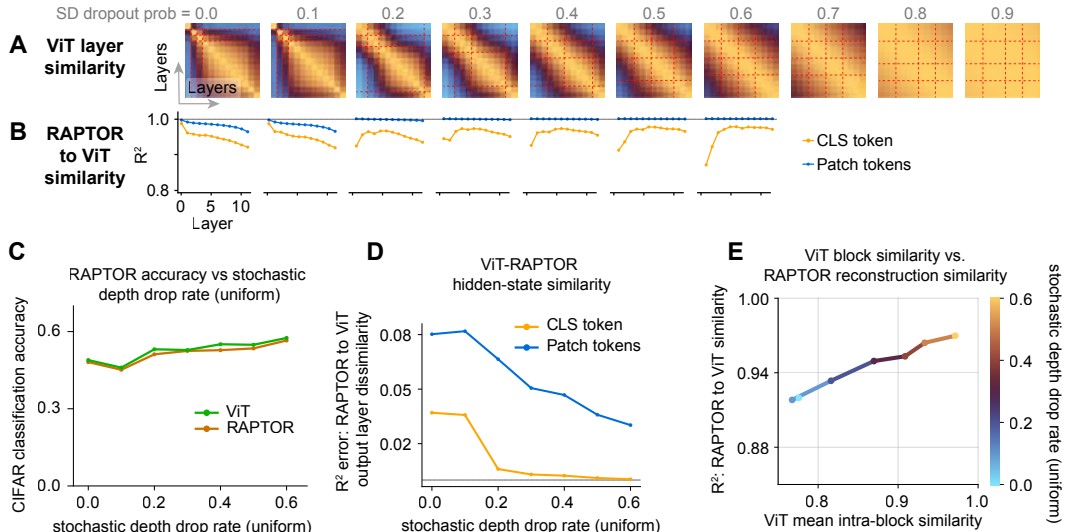

Figure 4: **Stochastic depth promotes representational similarity across layers block-recurrence.**
**A)** ViT layer-layer cosine similarity matrices for models trained with increasing stochastic depth
(SD) dropout probability $p$ (probabilities of 0.0-0.9, uniform over layer depth). Dashed red lines
delineate blocks, as defined by the max-cut algorithm. Higher SD $p$ values lead to a more similar
representation across layers. **B)** Layerwise teacher-student representational alignment $R^2$ (Raptor vs.
ViT) of the class `cls` and patch tokens. Increases in SD $p$ correspond to an increase in the ability
of Raptor to match the ViT's layerwise representations. ViT models for SD=0.7-0.9 show abberant
training dynamics and are excluded from this and further analysis. **C)** CIFAR classification accuracy
for a ViT trained on CIFAR, and a Raptor model with $k = 3$ blocks trained to match the hidden
state of the ViT. **D)** Last layer hidden-state similarity $R^2$ error (equivalent to $1 - R^2$) of the ViT
and Raptor model as a function of SD $p$. Increases in stochastic depth lead to a greater ability
to reconstruct ViT function using Raptor . **E)** Association between layer-layer representational
similarity and Raptor reconstruction $R^2$. Stochastic depth encourages the formation of more similar
blocks of layers within the ViT, which facilitates approximation by the recurrent Raptor model.

Taken together, these toy-model experiments support the view that the observed representational block
structure reflects both an intrinsic property of the residual ViT architecture and a learned/emergent
functional recurrence. Using the methods established here, we next scale up our application of
Raptor to modern large-scale foundation models.

## 3 SCALING Raptor TO FOUNDATION MODELS

Having demonstrated the BRH on controlled experiments, we now test whether it extends to large-
scale foundation models. We apply Raptor to DINOv2, chosen for its widespread adoption across
vision tasks, and optimize it to reproduce DINOv2's internal activations on ImageNet-1k.

**Architecture and Training.**    For all experiments, we use ImageNet-1K and extract activations from
a pretrained DINOv2 (ViT-Base) model, applying our max-cut algorithm to identify $k \in \{2, 3, 4\}$
recurrent block partitions. Each recurrent block $\boldsymbol{B}(\cdot)$ mirrors the block of DINOv2 (for detail about
the block see Section A). We train Raptor using a two-stage approach that combines teacher forcing
(TF) and autoregressive (AR) objectives. In the first stage, teacher forcing trains each block to predict
the immediate next layer given the correct previous layer, while the autoregressive objective requires
the model to use its own predictions as inputs for subsequent layers. Refer to Figure 11 for this hybrid
training algorithm. This approach yields the following total loss:

$$\mathcal{L}_{\text{total}}(\boldsymbol{x}) = \lambda \mathcal{L}_{\text{TF}}(\boldsymbol{x}) + (1 - \lambda)\mathcal{L}_{\text{AR},H}(\boldsymbol{x}) + \Omega(\boldsymbol{\theta}),$$

where $\Omega(\boldsymbol{\theta})$ denotes additional regularization applied to each tied block. See appendix A for
complete details. An attentive reader will notice that this training approach naturally lends itself
to parallelization: since each block operates on a distinct layer range, the first training stage can
be executed simultaneously across multiple GPUs or machines. Each block learns to approximate

its designated segment of the original network using the combined objective, with teacher forcing gradually annealed to zero as training progresses. The first stage thus allows blocks to develop their specific computational roles while benefiting from ground-truth activations as inputs.

The second stage connects all trained blocks into the complete recurrent architecture and trains the entire system end-to-end using only the autoregressive loss (i.e. $\lambda = 0$). This crucial phase teaches blocks to coordinate their computations and handle their own predicted activations rather than relying on ground-truth inputs from the teacher network. The transition from teacher forcing to pure autoregression ensures that the final model can operate independently while maintaining fidelity to the original network's representational trajectory.

We provide an implementation framework at `https://kempnerinstitute.github.io/raptor`. With the training methodology established, we now evaluate how effectively `Raptor`s can reproduce the performance of their teacher networks across multiple vision tasks.

**Results.** We evaluate `Raptor` against DINOv2 by training linear probes on ImageNet-1k (classification), ADE20k (semantic segmentation), and NYUv2 (monocular depth), covering both classification and dense prediction. For ImageNet-1k, we initialize the classifier from the public DINOv2 probe and report the best score across initialization and fine-tuning. In all experiments, the ViT backbone is frozen for both `Raptor` and DINOv2; only the linear heads are updated, and we reuse DINOv2's patch embedding and final layer normalization (both also frozen). Results appear in Table 2. `Raptor` performs well across tasks and is stronger on classification: with $k = 3$ it attains $83.0\%$ top-1 on ImageNet-1k (about $98\%$ of DINOv2 ViT-B and above ViT-S; see Fig. 5), with little deviation across runs ($\sigma = 0.1$). Accuracy improves markedly from $k = 2$ to $k = 3$ and then saturates at $k = 4$. In short, a two-block `Raptor` retains about $96\%$ of DINOv2 ViT-B with a frozen backbone, a compact rewriting that substantiates the BRH. This conclusion is further supported by Figure 14, which shows that `Raptor` maintains a high cosine similarity to `DINOv2` activations through depth, confirming that it captures the representational dynamics of the original model.

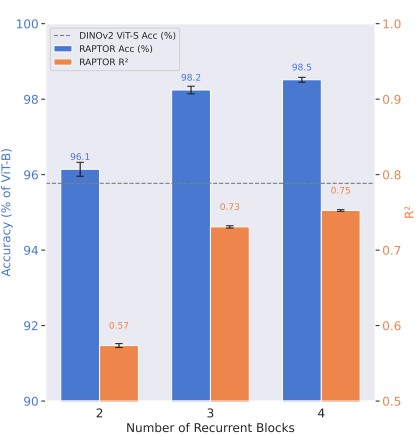

Figure 5: **`Raptor`'s performance on ImageNet-1k as a function of DINOv2 ViT-B accuracy (left), and $R^2$ score (right).** DINOv2 ViT-S accuracy shown as a dashed horizontal line. Results are aggregated over three model runs trained on different randoms seeds.

**Ablations.** Although our aim is not maximal compression nor exact accuracy matching, we perform targeted ablations to identify the factors most critical to `Raptor` performance (Table 1). Training with teacher forcing alone (Stage 1 only), while computationally efficient, leads to complete collapse with poor accuracy ($\sim 3\%$ on ImageNet-1K), indicating that one-step supervision is insufficient without exposure to the full autoregressive trajectory. Introducing the autoregressive loss and gradually annealing teacher forcing to zero raises accuracy by more than $68\%$, underscoring the necessity of closed-loop training for stable block-recurrent approximation. Further gains come from depth scaling (Appendix A.3), where each block has a learned vector embedding of its target layer index, making `Raptor` a non-autonomous dynamical

| Method | Accuracy |
|---|---|
| Teacher Forcing (TF) | 3.9 |
| + Autoreg (anneal TF) | 72.7 ↑ 68.8 |
| + Depth Scaling | 75.2 ↑ 2.5 |
| + Weighted cls | 76.7 ↑ 1.5 |
| + Second Stage | 82.4 ↑ 5.7 |
| + Finetune (Classifier) | 83.0 ↑ 0.6 |

Table 1: **Ablations to original `Raptor(k=3)` model**, showing ImageNet-1k accuracy with DINOv2 pretrained linear classifier. Second Stage refers to putting all three blocks together and training the full model autoregressively.

system (i.e., the update rule explicitly depends on the iteration count rather than state alone). Up-weighting the final block's cls token loss yields further improvements (Eq. 3, Appendix A). Finally, connecting all blocks and fine-tuning the model end-to-end with the autoregressive objective (i.e. the second stage) dramatically increases performance, and a final boost is obtained by fine-tuning the linear probe. Now that we have shown that the BRH holds for a foundation model, and before turning

| Method | Arch. | IN-1k (Acc ↑) | ADE20k (mIoU ↑) | NYUv2 (RMSE ↓) |
|---|---|---|---|---|
| Raptor | $k = 2$ | $81.2 \pm 0.2$ | $39.6 \pm 0.6$ | $0.648 \pm 0.003$ |
| | $k = 3$ | $83.0 \pm 0.1$ | $43.0 \pm 0.3$ | $0.618 \pm 0.006$ |
| | $k = 4$ | $83.2 \pm 0.1$ | $43.6 \pm 0.1$ | $0.607 \pm 0.006$ |
| DINOv2 | ViT-S | $80.9$ | $44.6$ | $0.600$ |
| | ViT-B | $84.5$ | $47.5$ | $0.578$ |

Table 2: **Performance of Raptor compared to DINOv2 with linear probes.** We report top-1 accuracy on ImageNet-1k, mean Intersection-over-Union (mIoU) on ADE20k semantic segmentation, and root mean squared error (RMSE) on NYUv2 depth estimation. Higher values are better for accuracy and mIoU, while lower values are better for RMSE. Results for Raptor are aggregated over three model runs, each trained with a different random seed, and displayed as $\mu \pm \sigma$. For Raptor, *Arch* denotes the number of recurrent blocks, while for DINOv2, *Arch* denotes the ViT backbone.

to Dynamical Interpretability, we first examine one implication of this phenomena: the algorithmic and computational implications of the BRH.

**Algorithmic and computational implications.** At scale, BRH holds in practice: a two-block Raptor recovers most of DINOv2 ViT-B, with three blocks essentially closing the gap. This reveals a strong simplicity bias in trained ViTs: depth reuses a small set of computations, effectively trading parameters for iterations. This reuse has two immediate consequences. First, it compresses the program description length that realizes the network's computation, suggesting low algorithmic complexity. However, the implication is more subtle than standard Kolmogorovcomplexity (Kolmogorov, 1965). While Kolmogorov compression can replace a long program with an arbitrarily short one that runs in unbounded time, Raptor crucially preserves computational cost: applying the same block $n_j$ times achieves equivalent runtime to $n_j$ distinct untied blocks. In other words, ViTs admit a more compact program representation *under the same runtime budget*, aligning more closely with Levin's complexity $K_{\text{Levin}}$ (Levin, 1973). At scale, BRH holds in practice: a two-block Raptor recovers most of DINOv2 ViT-B, and three blocks close the gap. This reveals a simplicity bias in trained ViTs: depth reuses computations, trading parameters for iterations.

**Claim 1** (BRH guarantees low Levin complexity)**.** *Let $\boldsymbol{f}_\ell$ satisfy 0-BRH with $k \ll \ell$ tied blocks $\{\boldsymbol{B}_j\}_{j=1}^k$ and schedule $(n_j)_{j=1}^k$ with $\sum_j n_j = \ell$. Let $R(\cdot)$ denote block runtime and define the untied teacher runtime $R(\boldsymbol{f}_\ell) := \sum_{j=1}^k n_j R(\boldsymbol{B}_f)$; assume runtime parity $R(\boldsymbol{B}_j) \leq (1 + \delta)R(\boldsymbol{B}_f)$. Then*

$$K_{\text{Levin}}(\boldsymbol{f}_\ell) \leq \sum_{j=1}^k DL(\boldsymbol{\theta}(\boldsymbol{B}_j)) + O(k \log \ell) + \log R(\boldsymbol{f}_\ell) + O(1).$$

See Appendix E for details. Theoretically, BRH guarantee low Levin complexity (compact algorithmic descriptions at unchanged computational cost). Our empirical validation of BRH on DINOv2 then suggest that foundation vision models are algorithmically simpler than their nominal architecture suggests. This compression reinforces emerging evidence that simplicity principles govern successful neural networks (Goldblum et al., 2023; Valle-Perez et al., 2019; Huh et al., 2023). For interpretability research, this offers an encouraging perspective: there exist representational lenses under which seemingly complex models reveal underlying simplicity. High-performing ViTs discover and iteratively reuse a compact set of algorithmic primitives, which is a structural regularity that may provide tractable entry points for mechanistic understanding.

We now pursue one such entry point. Since ViTs compress to recurrent blocks applied iteratively, we propose to analyze their computation as discrete-time dynamical systems, in the next section, we develop this *Dynamical Interpretability* framework to extract insights from the recurrent structure.

## 4 FROM BLOCK RECURRENCE TO *Dynamical Interpretability* IN VITS

Having observed block-structured representational similarity and confirmed that this similarity translates to functional recurrence, even in foundational models, we are inclined to seriously consider ViTs as dynamical systems that can be interpreted using dynamical systems analysis tools (Schmid, 2022; Ostrow et al., 2023; Huang et al., 2025) – what we term *dynamical interpretability*. We begin by establishing the basic dynamical properties of this depth flow, and present three key findings: (***i***) tokens converge directionally toward angular attractors with self-correcting dynamics, (***ii***) different token types exhibit specialized dynamics with punctuated transitions at phase boundaries, and (***iii***) later layers exhibit low-rank collective motion under weak contraction, reminiscent of mean-field processes with collapsing update dimensionality.

**Directional Convergence and Angular Attractor Geometry.** We begin by isolating direction and scale. Feature norms increase steadily across depth, precluding Euclidean notions of convergence or attractors; we therefore normalize states and study their angular evolution (Fig. 12). Concretely, let

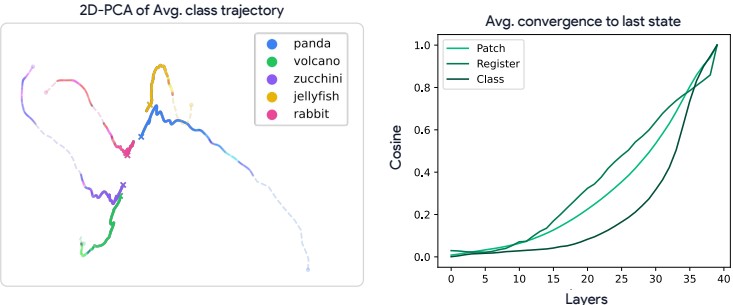

Figure 6: **Directional convergence on the unit sphere. (Left)** Average normalized trajectories (PCA space) collapse into compact class-dependent basins, consistent with low-dimensional angular attractors. **(Right)** Cosine similarities to final representations $\gamma_\ell$ are S-shaped, saturating near 1 for `cls`, `registers`, and `patch`, indicating directional fixed points.

$\hat{x}_\ell = x_\ell/\|x_\ell\|$ denote the direction of a token at layer $\ell$, and consider the depth trajectory $\{\hat{x}_\ell\}_{\ell=0}^L$ on the unit sphere $\mathbb{S}^{d-1}$. Directional convergence is quantified by $\gamma_\ell = \langle \hat{x}_\ell, \hat{x}_L \rangle$. Empirically, $\gamma_\ell$ follows smooth S-shaped curves that approach 1 and saturate in late layers for all token types (Fig. 6, right). This behavior indicates a directional fixed point: while norms grow, directions stabilize so that $\hat{x}_{\ell+1} \approx \hat{x}_\ell$ as $\ell$ increases. The late acceleration of $\gamma_\ell$ suggests phase-local attraction strengthening in the final phase, clarified by our coherence study (Fig. 9, middle). A complementary geometric view comes from projecting the depth trajectories onto a low-dimensional subspace. PCA reveals that sample-specific paths enter class-dependent basins in a shared angular subspace; across $1,000$ images from 5 imagenet classes, trajectories curl into compact terminal regions rather than scattering (Fig. 6, left). We interpret these regions as angular attractors: small sets on $\mathbb{S}^{d-1}$ toward which iterates of the phase-local map steer directions, up to within-class variability. Finally, we probe stability by injecting a small additive perturbation at layer $\ell$ and following the perturbed direction thereafter. The average perturbed path bends back toward the unperturbed trajectory, indicating local self-correction and on-sphere contraction around the limiting direction (Fig. 8). Together, these measurements establish property (i): token directions evolve under depth toward angular attractors with mild contraction, making directional geometry an appropriate lens for subsequent dynamical analysis.

**Token-Specific Dynamics.** Token groups follow distinct angular update laws. For a token with normalized state $\hat{x}_\ell$, define the per-layer angular speed $s_\ell = \arccos\langle \hat{x}_{\ell+1}, \hat{x}_\ell \rangle$. Aggregating $s_\ell$ by token type reveals stable small speeds for registers, intermediate speeds for patches, and sharp late reorientations for `cls` (Fig. 7). The variance of $s_\ell$ is smallest for registers after early depth; conversely, `cls` shows increased late angular activity, reflecting its global aggregator function. These token-specific laws vary across depth. Angular speed statistics display abrupt changes aligned with discovered block boundaries, creating a punctuated pattern where phases maintain near-stationary behavior that is reset at phase transitions (Fig. 7). This structure matches the block-recurrent view in which a phase applies a reused update map with stable statistics before handing off to a new regime at the boundary. Sensitivity analyses corroborate this specialization. Inject a small additive perturbation of magnitude $\varepsilon$ at layer $\ell$ and measure the final angular deviation using the cosine distance $d_{\cos}(\hat{x}_L^{(\varepsilon,\ell)}, \hat{x}_L)$. The scaled sensitivity $|\varepsilon|^{-1}d_{\cos}$ decays

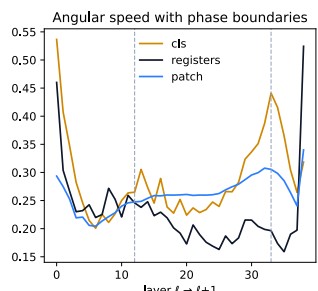

Figure 7: **Token-specific angular speed with phase overlays.** Mean angular speed $s_\ell$ across depth for `cls`, registers, and `patch`, with max-cut phase boundaries from Sec. 2 overlaid as vertical lines.

approximately log-linearly with deeper injection for patch tokens, indicating on-sphere attenuation within phases, whereas it increases for `cls` when injected late, indicating accumulation at the read-out stage where global information is consolidated (Fig. 8B). Directional convergence rates mirror these roles. Tracking $c_\ell = \langle \hat{x}_\ell, \hat{x}_L \rangle$ reveals registers approach their terminal directions earliest, patches follow with a smoother rise, and `cls` saturates only in the final phase where its reorientation peaks (Fig. 6, left). Together, these measurements demonstrate ViT depth implements specialized, phase-local dynamics with phase transitions, consistent with block-recurrent computation.

**Low-Rank Collective Motion and Linearized Depth Flow.** We analyze layer-to-layer token-wise angular updates and observe a progressive collapse to low dimensions. For token-wise angular updates (see App. C). Stable rank and effective rank decrease with depth to near six in the final phase, indicating subspace confinement (Fig. 9, left). Concurrently, patch-token coherence $\kappa_\ell$ rises sharply

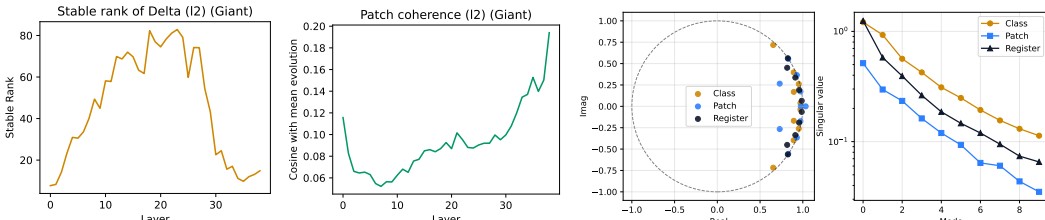

Figure 8: **Self-correction under small angular perturbations.** Perturbed trajectories bend back toward the baseline path, evidencing local basin stability. Sensitivity decays approximately log-linearly with remaining depth for `patch` tokens, but grows late for `cls`, consistent with stronger late-stage aggregation.

Figure 9: **(Left) Low-rank updates and coordinated patch motion.** Left Stable and effective rank of the layer-to-layer update matrix collapse with depth, indicating confinement to a restricted subspace. Right Patch-token coherence with their mean update direction rises strongly, revealing increasing collective alignment. **(Right) Dynamic Mode Decomposition (DMD) of depth dynamics.** For each token group (`cls`, registers, `patch`), we average token states within the group and fit the exact-DMD (see Section D). Each layer state is $\ell_2$-normalized to unit norm (trajectories on the unit sphere), so eigenvalue angles $\arg(\lambda_i)$ characterize angular updates, while radii $|\lambda_i|$ measure contraction on the sphere (not absolute feature-norm growth). The DMD eigenvalues $\{\lambda_i\}$ lie just inside the unit circle (dashed) and concentrate near the positive real axis, indicating near-neutral, mostly angular updates with mild on-sphere contraction. `cls` modes lie closest to $+1$ (longest memory), registers are slightly more dispersed, and `patch` shows the widest angular spread and stronger contraction. The `cls` spectrum decays slowest (highest effective rank/complexity), registers are intermediate, and `patch` decays fastest (lower-rank dynamics). Together, these spectra support a weakly contracting, block-recurrent depth flow with token-specific complexity.

and peaks late, showing increasingly aligned, collective updates (Fig. 9, middle). The joint pattern (rank collapse with rising coherence) marks a transition from many weakly independent directions to a few shared directions. We then linearize the depth flow via exact DMD on group-averaged, $\ell_2$-normalized states, yielding $\bar{x}_{\ell+1} \approx A\bar{x}_\ell$ with rank $r = 10$ (App. D). Eigenvalues are concentrated just inside the unit circle and near the positive real axis, consistent with weak on-sphere contraction and predominantly angular updates; `cls` modes lie closest to $+1$ (longest memory), registers are intermediate, and patches show wider angular spread and stronger contraction (Fig. 9, right). Stacked-depth singular spectra mirror this ordering, decaying slowest for `cls` and fastest for patches. These results indicate that late depth implements low-rank, near-neutral dynamics that compress variation into a small set of collective directions while preserving long-memory channels for `cls`.

## 5 DISCUSSION

We advanced the Block-Recurrent Hypothesis (BRH), showing empirically and constructively (via weight-tied surrogates) that recurrence can match untied baselines, and we developed *Dynamical Interpretability* by viewing depth as a flow on directions. This revealed (***i***) directional convergence to angular attractors with self-correction, (***ii***) token-specific, phase-local dynamics with punctuated transitions, and (***iii***) a late low-rank regime that coordinates updates to low dimensional subspace. While residual pathways and stochastic depth appear implicated in block recurrence, isolating causal mechanisms will require controlled training-dynamics at scale; and although two tied blocks recover most of DINOv2, a small residual gap remains that may call for improved recurrent distillation or additional time-varying components. Overall, our work highlights a recurrence induced simplicity bias, suggesting current models admit a recurrent version, implicating a potential simpler analysis. Taken together, this recurrence-induced simplicity bias and its interpretability potential point toward a broader principle: in deep learning, *recurrence finds a way*.

## 6 ACKNOWLEDGMENTS

The authors would like to thank Chris Hamblin, Louis Béthune and John Hammond for many fruitful discussions. This work has been made possible in part by a gift from the Chan Zuckerberg Initiative Foundation to establish the Kempner Institute for the Study of Natural and Artificial Intelligence at Harvard University. MJ is supported by the Kempner Institute Graduate Research Fellowship. TAK, TF, and RH are supported by the Kempner Institute Research Fellowship. DB is supported by the Kempner Institute for the Study of Natural and Artifical Intelligence at Harvard University.

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

# A  TRAINING BLOCK RECURRENT FOUNDATION MODELS

All `Raptor` variants ($k = 2, 3, 4$) are trained on top of DINOv2 (ViT-B with registers) (Darcet et al., 2023) and use the same transformer architecture. The model utilizes a feature dimension of 784, an MLP ratio of 4, and 12 multi-head attention heads. The depth-scale MLP in each block generates three separate scaling vectors; we provide the full formulation in Appendix A.3. For each `Raptor` variant in Table 2 and Figure 5, we train train three models, each with a different seed. We measure the performance of each model and report aggregate statistics. Below, we provide the hyperparameters and training settings used to train `Raptor`, describing the layer divisions for different values of $k$ followed by the training procedure.

For each choice of $k$, the encoder layers are divided into blocks as shown in Table 3. We select these partitions using the max cut algorithm applied to the DINOv2 ViT-B activation layer-layer cosine similarity matrix using 10000 samples from the ImageNet-1k training set.

| $k$ | Block | Input $\rightarrow$ Predicted Layers |
|---|---|---|
| 2 | Block 1 | $0 \rightarrow 1$–$7$ |
|   | Block 2 | $7 \rightarrow 8$–$12$ |
| 3 | Block 1 | $0 \rightarrow 1$–$7$ |
|   | Block 2 | $7 \rightarrow 8$–$10$ |
|   | Block 3 | $10 \rightarrow 11$–$12$ |
| 4 | Block 1 | $0 \rightarrow 1$–$4$ |
|   | Block 2 | $4 \rightarrow 5$–$7$ |
|   | Block 3 | $7 \rightarrow 8$–$10$ |
|   | Block 4 | $10 \rightarrow 11$–$12$ |

Table 3: Layer splits used for training with different values of $k$.

In the first stage, each block is trained independently on the subset of layers it is responsible for predicting. For example, when $k = 3$, Block 1 is trained to predict Layers 1–7. We train on the ImageNet-1k train split for 20 epochs with a batch size of 64 using the AdamW optimizer with a weight decay of 0.0001. The learning rate follows a linear warmup for 10,000 steps to $1 \times 10^{-4}$, followed by a cosine decay to $1 \times 10^{-6}$. The teacher forcing loss weight $\lambda$ is annealed from 0.5 to 0 over the first 5 epochs. For the third block, we use specific token loss weights of $\lambda_{\texttt{cls}} = 0.34$, $\lambda_{\texttt{reg}} = 0.33$, and $\lambda_{\texttt{patch}} = 0.33$.

In the second stage, after independent training, all blocks are connected to autoregressively predict Layers 1–12 end-to-end. Each block still predicts its designated segment, but the entire model now backpropagates through the full sequence. We maintain the same dataset, epoch count, batch size, weight decay, optimizer, and learning rate schedule as the first stage. However, the token loss weights are adjusted to $\lambda_{\texttt{cls}} = 0.45$, $\lambda_{\texttt{reg}} = 0.1$, and $\lambda_{\texttt{patch}} = 0.45$.

The number of recurrent timesteps each block executes corresponds directly to its layer coverage; for instance, in the k=3 setup, Block 1 runs for 7 timesteps, Block 2 for 3 timesteps, and Block 3 for 2 timesteps.

For a given batch of ground truth DINOv2 activations $\boldsymbol{X} \in \mathbb{R}^{N \times T \times D}$ and `Raptor` predictions $\hat{\boldsymbol{X}} \in \mathbb{R}^{N \times T \times D}$, where $N$ denotes the batch size, $T$ the number of tokens, and $D$ the feature dimension.

We decompose the activation tensor along the token dimension into three distinct components corresponding to the token types: the class token $\boldsymbol{X}_{\texttt{cls}} \in \mathbb{R}^{N \times 1 \times D}$, the register tokens $\boldsymbol{X}_{\texttt{reg}} \in \mathbb{R}^{N \times N_{\texttt{reg}} \times D}$, and the spatial patch tokens $\boldsymbol{X}_{\texttt{patch}} \in \mathbb{R}^{N \times N_{\texttt{patch}} \times D}$, such that $\boldsymbol{X} = \text{concat}(\boldsymbol{X}_{\texttt{cls}}, \boldsymbol{X}_{\texttt{reg}}, \boldsymbol{X}_{\texttt{patch}})$.

We define the total reconstruction loss function $\mathcal{L}(\boldsymbol{X}, \hat{\boldsymbol{X}})$ as a weighted sum of the errors for each component:

$$\mathcal{L}(\boldsymbol{X}, \hat{\boldsymbol{X}}) = \lambda_{\texttt{cls}} \|\hat{\boldsymbol{X}}_{\texttt{cls}} - \boldsymbol{X}_{\texttt{cls}}\|_F^2 + \lambda_{\texttt{reg}} \|\hat{\boldsymbol{X}}_{\texttt{reg}} - \boldsymbol{X}_{\texttt{reg}}\|_F^2$$
$$+ \lambda_{\texttt{patch}} \|\hat{\boldsymbol{X}}_{\texttt{patch}} - \boldsymbol{X}_{\texttt{patch}}\|_F^2, \tag{3}$$

### A.1 PHASE DISCOVERY VIA A CONTIGUOUS MAX-CUT ON THE LAYER–LAYER SIMILARITY

**Problem setup.** Let $S \in \mathbb{R}^{L \times L}$ be the (symmetrized) layer-layer similarity matrix, where $S_{ij}$ measures the similarity between layers $i$ and $j$ (for example, cosine similarity). We seek a partition of depth into $k$ contiguous segments or "phases". $\Pi = \{[b_1, e_1], \ldots, [b_k, e_k]\}$ with $1 = b_1 \leq e_1 < b_2 \leq e_2 < \cdots < b_k \leq e_k = L$ and $e_t + 1 = b_{t+1}$, that maximizes within-block similarity (equivalently, minimizes cross-block cut).

**Objectives.** For a segment $[i, j]$ of length $n = j - i + 1$, define:

$$\text{sum}(i, j) = \sum_{p=i}^{j} \sum_{q=i}^{j} S_{pq}, \quad \text{offdiag}(i, j) = \text{sum}(i, j) - \sum_{p=i}^{j} S_{pp}.$$

We consider additive segment scores $g(i, j)$ by computing the final weighted mean as:

$$\frac{\text{sum}(i, j)}{n^2};$$

Maximizing $\sum_{t=1}^{k} g(b_t, e_t)$ prefers blocks that are internally similar and, by contiguity, implies small cross-block interfaces (a contiguous max-cut on the line).

**Fast block queries via 2-D prefix sums.** Precompute a 2-D prefix (summed-area) table $P \in \mathbb{R}^{(L+1) \times (L+1)}$ with $P_{rc} = \sum_{u<r} \sum_{v<c} S_{uv}$. Then any submatrix sum obeys

$$\text{sum}(i, j) = P_{j+1,j+1} - P_{i,j+1} - P_{j+1,i} + P_{i,i},$$

in $O(1)$ time; diagonal sums use a 1-D prefix over $\text{diag}(S)$. This is sometimes referred to as the "integral image" trick.

**Contiguous DP solver** $(O(kL^2))$. Let $\text{dp}[t, j]$ be the best score for partitioning layers $1..j$ into $t$ blocks. With minimum block length $m$,

$$\text{dp}[1, j] = g(1, j) \quad (j \geq m), \qquad \text{dp}[t, j] = \max_{i \in \{t\,m-1, \ldots, j-m\}} \text{dp}[t - 1, i] + g(i + 1, j),$$

for $t = 2, \ldots, k$ and $j \geq t\,m$. We keep backpointers $\text{prev}[t, j]$ to recover boundaries by backtracking from $(t=k, j=L)$. With $g(\cdot)$ evaluated in $O(1)$ by prefix sums, the overall complexity is $O(kL^2)$ time and $O(kL)$ memory. This DP structure mirrors classical optimal 1-D segmentation/partitioning solvers.

### A.2 TEACHER-STUDENT RECONSTRUCTION $R^2$

To stabilize measures of pairwise vector similarity over large hyperparameter sweeps when fits may be poor, we use an alternative calculation for $R^2$ for small-scale models (Figure 4). Here, we first regress the student's `cls` or `patch` tokens $\in R^{N \times D}$ to the corresponding teacher tokens $\in R^{N \times D}$ using ordinary least squares with a bias term. $N$ is the number of tokens and $D$ is the dimensionality of the token. We then calculate the average of the $R^2$ values between the true teacher token vectors and the student's reconstruction of those vectors. Note that this regression is purely a 1-dimensional rescaling and shifting for each student token vector. This results in an $R^2$ value that is bounded between 0 and 1, and can be understood to present the 'explainable variance' between the student and teacher representations.

### A.3 DEPTH-SCALING MECHANISM

We employ a lightweight conditioning MLP in each block that maps a scalar depth coordinate $z \in \mathbb{R}$ to layer-specific scaling vectors. For a layer with feature dimension $D = 784$, the depth-scale MLP generates a modulation vector $\mathbf{v} \in \mathbb{R}^{3D}$.

The MLP projects the scalar input to a hidden dimension $h = 16$, applies a SiLU activation, and projects to the output dimension. We add a unity offset to the MLP output to ensure the scaling factors center around 1 (identity behavior) at initialization. The compound scaling vector $\mathbf{S}$ is computed as:

$$\mathbf{S} = (\mathbf{W}_2 \cdot \mathrm{SiLU}(\mathbf{W}_1 z + \mathbf{b}_1) + \mathbf{b}_2) + \mathbf{1} \tag{4}$$

where $\mathbf{1}$ denotes a vector of ones. We initialize the weights and biases of the final projection $(\mathbf{W}_2, \mathbf{b}_2)$ with a normal distribution ($\mu = 0, \sigma = 10^{-4}$), ensuring that the block initially acts close to standard behavior.

The vector $\mathbf{S}$ is split along the feature dimension into three component vectors: $\mathbf{s}_{\mathrm{attn}}, \mathbf{s}_{\mathrm{mlp}}, \mathbf{s}_{\mathrm{out}} \in \mathbb{R}^D$. These vectors modulate the signal via element-wise multiplication ($\odot$) at three points in the transformer block: the attention residual, the MLP residual, and the final block output. Given an input sequence $\mathbf{X}$, the forward pass is defined as:

$$\mathbf{X}' = \mathbf{X} + \mathbf{s}_{\mathrm{attn}} \odot \mathrm{LS}_1(\mathrm{Attn}(\mathrm{LN}_1(\mathbf{X}))) \tag{5}$$

$$\mathbf{X}'' = \mathbf{X}' + \mathbf{s}_{\mathrm{mlp}} \odot \mathrm{LS}_2(\mathrm{MLP}(\mathrm{LN}_2(\mathbf{X}'))) \tag{6}$$

$$\mathbf{Y} = \mathbf{s}_{\mathrm{out}} \odot \mathbf{X}'' \tag{7}$$

where LS refers to LayerScale, LN is Layer Normalization, and broadcasting is applied over the sequence length dimension.

### A.4 LINEAR PROBE FINE-TUNING

We fine-tune linear probes on three downstream datasets: ImageNet-1k (classification), ADE20k (semantic segmentation), and NYUv2 (monocular depth estimation).

For ImageNet-1k and ADE20k, we use the AdamW optimizer with linear warmup followed by cosine learning rate decay. For NYUv2, we use AdamW with `GradScaler` and mixed precision training. All probes operate on the final block's prediction of Layer 12, using either the `cls` token, patch tokens, or both, depending on the task. The detailed hyperparameters are shown in Table 4.

For NYUv2, we adopt an approach similar to Oquab et al. (2023). Specifically, we use images at a $480 \times 640$ resolution and center pad them so that the dimensions are multiples of 14. We feed the images through the model and extract the predictions from the final layer. The `cls` token is concatenated with the patch tokens, and the spatial resolution is upsampled by a factor of 4. Both the `cls` and patch tokens are upscaled, after which the `cls` token is concatenated to each patch token. We treat this representation as the "logits." To obtain depth, we normalize the logits with a softmax and compute the weighted average of the centers of 256 evenly spaced bins. Then, we upsample this representation to $480 \times 640$ and consider the result our depth. For training, we use the loss function introduced by Bhat et al. (2021).

| Hyperparameter | ImageNet-1k | ADE20k | NYUv2 |
|---|---|---|---|
| Epochs | 15 | 10 | 25 |
| Batch Size | 512 | 32 | 128 |
| Base LR | $1 \times 10^{-3}$ | $1 \times 10^{-3}$ | $1 \times 10^{-4}$ |
| Weight Decay | $1 \times 10^{-2}$ | $1 \times 10^{-2}$ | $1 \times 10^{-2}$ |
| Grad. Clip Norm | 1.0 | 1.0 | 1.0 |
| Warmup Iters | 100 | 100 | 100 |
| Optimizer | AdamW | AdamW | AdamW + GradScaler |
| Head Init. | DINOv2 classification probe | Random segmentation head | Random depth head |
| Input Tokens | concat(`cls`, mean patch) | Patch | concat(`cls`, patch) |

Table 4: Linear probe fine-tuning hyperparameters across datasets. Base LR denotes the peak learning rate before cosine decay.

## B EVALUATING MAX-CUT ALGORITHM

To evaluate the efficacy of the max cut algorithm (Figure 3), we initialize a ViT-B with ImageNet-21k and ImageNet-1k weights and replace the classifier with a randomized linear probe applied

to the CLS token. We fine-tune the full model on CIFAR-100 for 10 epochs, achieving $90.7\%$ validation accuracy. Using activations extracted from the CIFAR-100 training set, we compute a layer-wise cosine similarity matrix and apply our max cut algorithm to identify `Raptor` partitions for $k \in \{2, 3, 4\}$. We compare these against two baselines: 10 random contiguous partitions and 10 random non-contiguous partitions (labeled "Random Shuffle"), ensuring the max-cut solution is excluded from the samples. Finally, we train `Raptor` models on each partition configuration for 100 epochs and evaluate performance using the linear probe from the initial fine-tuning stage.

## C DYNAMICS PROTOCOLS AND METRICS

This appendix consolidates definitions and experimental procedures used in Sec. 4. All measurements are performed on ImageNet validation data. For aggregate statistics, we use 10k randomly sampled validation images. For trajectory visualizations (e.g., Fig. 6), we select five ImageNet classes with 1,000 images each. Inputs are resized to 256 pixels on the shorter side and center-cropped to $224 \times 224$. Unless otherwise noted, we use `DINOv2-Giant` with four register tokens from the official implementation.

**Normalization.** Token states $\boldsymbol{x}_\ell \in \mathbb{R}^d$ at depth $\ell$ are decomposed into norm and direction. We study normalized states

$$\hat{\boldsymbol{x}}_\ell = \frac{\boldsymbol{x}_\ell}{\|\boldsymbol{x}_\ell\|} \in \mathbb{S}^{d-1},$$

so that dynamics are restricted to the unit sphere.

**Directional convergence.** Directional similarity to the terminal representation is measured by

$$\gamma_\ell = \langle \hat{\boldsymbol{x}}_\ell, \hat{\boldsymbol{x}}_L \rangle,$$

which traces the angular alignment of layer $\ell$ to the final state.

**Angular speed.** Per-layer angular update magnitude is defined as

$$s_\ell = \arccos\langle \hat{\boldsymbol{x}}_{\ell+1}, \hat{\boldsymbol{x}}_\ell \rangle.$$

Statistics of $s_\ell$ are stratified by token type.

**Phase overlays.** Phase boundaries are obtained from the max-cut segmentation of representational similarity matrices (Sec. 2) and used as vertical markers in angular speed and sensitivity plots.

**Perturbation protocol.** To probe stability, we add a perturbation $\varepsilon \boldsymbol{u}$ at layer $\ell$,

$$\tilde{\boldsymbol{x}}_\ell = \boldsymbol{x}_\ell + \varepsilon \boldsymbol{u}, \quad \boldsymbol{u} \sim \mathcal{N}(0, I_d),$$

and follow the normalized trajectory thereafter. Sensitivity is quantified by the terminal cosine deviation

$$d_{\cos}(\hat{\boldsymbol{x}}_L^{(\varepsilon, \ell)}, \hat{\boldsymbol{x}}_L) = 1 - \langle \hat{\boldsymbol{x}}_L^{(\varepsilon, \ell)}, \hat{\boldsymbol{x}}_L \rangle.$$

**Low-rank and coherence metrics.** For angular updates $\Delta_\ell^{(i)} = \hat{\boldsymbol{x}}_{\ell+1}^{(i)} - \hat{\boldsymbol{x}}_\ell^{(i)}$, we form the update matrix $\boldsymbol{U}_\ell$. Stable rank is given by

$$r_s(\boldsymbol{U}_\ell) = \frac{\|\boldsymbol{U}_\ell\|_F^2}{\|\boldsymbol{U}_\ell\|_2^2},$$

and coherence by

$$\kappa_\ell = \frac{1}{N} \sum_i \frac{\langle \Delta_\ell^{(i)}, \bar{\Delta}_\ell \rangle}{\|\Delta_\ell^{(i)}\| \, \|\bar{\Delta}_\ell\|}, \quad \bar{\Delta}_\ell = \frac{1}{N} \sum_i \Delta_\ell^{(i)}.$$

## D DYNAMIC MODE DECOMPOSITION

Let $\boldsymbol{f}$ be a trained ViT with transformer layers $\{\boldsymbol{f}_\ell\}_{\ell=1}^L$. For $\boldsymbol{x} \in \mathcal{X}$, denote by $\boldsymbol{A}_\ell(x) \in \mathbb{R}^{T \times d}$ the token matrix at depth $\ell$ with $T = 1 + R + P$ (cls, $R$ registers, $P$ patch). Form group states by

within-layer averaging

$$\boldsymbol{z}_\ell^{(\text{cls})}(x) = \boldsymbol{A}_\ell(x)_{\text{cls}} \quad \boldsymbol{z}_\ell^{(\text{reg})}(x) = \frac{1}{R} \sum_{t \in \mathcal{T}_{\text{reg}}} \boldsymbol{A}_\ell(x)_t \quad \boldsymbol{z}_\ell^{(\text{patch})}(x) = \frac{1}{P} \sum_{t \in \mathcal{T}_{\text{patch}}} \boldsymbol{A}_\ell(x)_t$$

and enforce per-layer $\ell_2$ normalization on the group averages

$$\boldsymbol{x}_\ell^{(g)}(x) = \frac{\boldsymbol{z}_\ell^{(g)}(x)}{\|\boldsymbol{z}_\ell^{(g)}(x)\|_2} \in \mathbb{S}^{d-1} \subset \mathbb{R}^d.$$

All DMD fits below are performed independently for each $g \in \{\text{cls}, \text{reg}, \text{patch}\}$ on the depth trajectory $\boldsymbol{x}_{0:L}^{(g)}(x)$. We start by stacking states along depth to form

$$\boldsymbol{Y}^{(g)} = \begin{bmatrix} (\boldsymbol{x}_0^{(g)})^\top \\ \vdots \\ (\boldsymbol{x}_L^{(g)})^\top \end{bmatrix} \in \mathbb{R}^{(L+1) \times d} \quad \boldsymbol{X}_1 = (\boldsymbol{Y}_{0:L}^{(g)})^\top \in \mathbb{R}^{d \times L} \quad \boldsymbol{X}_2 = (\boldsymbol{Y}_{1:L+1}^{(g)})^\top \in \mathbb{R}^{d \times L}.$$

DMD fits a single linear depth-step map $\boldsymbol{A}$ with $\boldsymbol{X}_2 \approx \boldsymbol{A}\boldsymbol{X}_1$. For the exact DMD at rank $r$, we compute the SVD $\boldsymbol{X}_1 = \boldsymbol{U}\boldsymbol{\Sigma}\boldsymbol{V}^\top$ and select $r \leq \text{rank}(\boldsymbol{X}_1)$. Let $\boldsymbol{U}_r\,\boldsymbol{\Sigma}_r\,\boldsymbol{V}_r$ be the leading blocks and define the reduced operator

$$\widetilde{\boldsymbol{A}} = \boldsymbol{U}_r^\top \boldsymbol{X}_2 \boldsymbol{V}_r \boldsymbol{\Sigma}_r^{-1} \in \mathbb{R}^{r \times r}.$$

Diagonalize $\widetilde{\boldsymbol{A}}\boldsymbol{W} = \boldsymbol{W}\boldsymbol{\Lambda}$ with $\boldsymbol{\Lambda} = \text{diag}(\lambda_1, \ldots, \lambda_r) \in \mathbb{C}^{r \times r}$ and $\boldsymbol{W} \in \mathbb{C}^{r \times r}$. The exact DMD modes in ambient space are

$$\boldsymbol{\Phi} = \boldsymbol{X}_2 \boldsymbol{V}_r \boldsymbol{\Sigma}_r^{-1} \boldsymbol{W} \in \mathbb{C}^{d \times r}.$$

Modal amplitudes for the initial state are $\boldsymbol{b} = \boldsymbol{\Phi}^\dagger \boldsymbol{x}_0^{(g)} \in \mathbb{C}^r$. One-step and $t$-step reconstructions are

$$\widehat{\boldsymbol{x}}_1^{(g)} \approx \boldsymbol{\Phi}\boldsymbol{\Lambda}\boldsymbol{b} \quad \widehat{\boldsymbol{x}}_t^{(g)} \approx \boldsymbol{\Phi}\boldsymbol{\Lambda}^t\boldsymbol{b} \text{ for } t \geq 0$$

with the induced linear predictor $\boldsymbol{A} = \boldsymbol{X}_2 \boldsymbol{V}_r \boldsymbol{\Sigma}_r^{-1} \boldsymbol{U}_r^\top$. No affine offset is fitted.

**On-sphere interpretation.** Because each $\boldsymbol{x}_\ell^{(g)}$ lies on $\mathbb{S}^{d-1}$ the map $\boldsymbol{A}$ is a best linear approximation of the depth flow restricted to the unit sphere. For $\lambda_i = |\lambda_i|e^{\mathrm{i}\theta_i}$ the modulus $|\lambda_i|$ measures contraction of directions within the span of modes on the sphere and the angle $\theta_i$ captures per-layer rotational change. The spectral radius $\rho(\boldsymbol{A}) = \max_i |\lambda_i|$ and the median of $|\lambda_i|$ summarize contraction strength. In particular this explain why all the eigenvalue in Figure 9 are contain in $\mathbb{S}^2$, see the report of the eigenvalue cloud $\{\lambda_i\}_{i=1}^r$ in the complex plane and the singular spectrum $\{\sigma_i\}_{i=1}^r$ of $\boldsymbol{X}_1$.

## E    LEVIN COMPLEXITY UNDER 0-BRH

**Background & Positioning.** There is a long line of work linking plain Kolmogorov complexity Kolmogorov (1965); Chaitin (1977) to its resource-bounded variants, notably Levin's time-bounded measure, which penalizes programs by both description length and runtime (the $|p| + \log T$ form) (Levin, 1973; Li et al., 2008). Deep learning research has not remained isolated from these ideas: connections have been explored both for discovering simpler neural networks Schmidhuber (1997) and for contextualizing runtime priors Schmidhuber (2002) within neural architectures.

In our case, the proof is a direct application of this standard framework as formalized in Li et al. (2008): once we exhibit a prefix-free program $p$ that computes $\boldsymbol{f}_\ell$ and bound its running time $T(p)$, Levin complexity immediately yields $K_{\text{Levin}}(\boldsymbol{f}_\ell) \leq |p| + \log T(p) + O(1)$. The only domain-specific ingredients are *(i)* a BRH-based encoding that prices the tied blocks and schedule at $\sum_{j=1}^k DL_U(\boldsymbol{\theta}(\boldsymbol{B}_j)) + O(k \log \ell)$ using standard self-delimiting integer codes, and *(ii)* a runtime-parity assumption linking $T(p)$ to the model's untied runtime $R(\boldsymbol{f}_\ell)$ up to machine and invariance constants. Formally:

**Claim 2** (BRH guarantees low Levin complexity). *Let $\boldsymbol{f}_\ell$ satisfy 0-BRH with $k \ll \ell$ tied blocks $\{\boldsymbol{B}_j\}_{j=1}^k$ and schedule $(n_j)_{j=1}^k$ with $\sum_j n_j = \ell$. Let $R(\cdot)$ denote block runtime and define the untied*

*teacher runtime* $R(\boldsymbol{f}_\ell) := \sum_{j=1}^{k} n_j R(\boldsymbol{B_f})$; *assume runtime parity* $R(\boldsymbol{B}_j) \le (1+\delta) R(\boldsymbol{B_f})$. *Then*

$$K_{\text{Levin}}(\boldsymbol{f}_\ell) \le \sum_{j=1}^{k} DL(\boldsymbol{\theta}(\boldsymbol{B}_j)) + O(k \log \ell) + \log R(\boldsymbol{f}_\ell) + O(1).$$

*Proof.* We follow the standard time-bounded description framework of Levin (1973), and contextualized it to the 0-BRH setting (perfect reconstruction along the entire depth trajectory), in the spirit of the sketch already outlined in the paper. We fix a universal prefix-free machine $U$ and adopt the same finite-precision arithmetic used by the target model to define $DLU(\cdot)$; this ensures that encoding parameters with length $DLU(\boldsymbol{\theta}(\boldsymbol{B}_j))$ bits suffices to reproduce the exact numerical behavior of the corresponding block under that arithmetic. With this convention, "exactly computes" means bit-for-bit equivalence with the reference implementation of $\boldsymbol{f}_\ell$ (under its inference precision).

The proof proceeds in three steps: we first bound the description length, then verify correctness (equality of the program output), and finally bound the runtime before combining these results to derive the final bound.

The first step consists in bounding the description length. We describe a prefix-free program $p$ that parses (i) the tied block parameters and (ii) a self-delimiting schedule, with total length

$$|p| \le \sum_{j=1}^{k} DLU(\boldsymbol{\theta}(\boldsymbol{B}_j)) + O(k \log \ell) + O(1).$$

For each tied block, encode its parameters using the scheme underlying $DLU$, yielding a prefix-free code of length $DLU(\boldsymbol{\theta}(\boldsymbol{B}_j))$. Concatenate the $k$ codes and prepend an $O(1)$ header that instructs $U$ how to parse and reconstruct each $\boldsymbol{B}_j$ from its bitstream. Encode the schedule $(n_1, \ldots, n_k)$ and the value of $k$ as self-delimiting integers (any standard universal code suffices), which costs $O(k \log \ell)$ bits because $n_j \le \ell$ and $\sum_j n_j = \ell$. The parser is a constant-size routine bundled with $p$, absorbed into $O(1)$.

Now we show that this construction is a valid under 0-BRH. Let $g_\ell := \boldsymbol{B}_k^{(n_k)} \circ \cdots \circ \boldsymbol{B}_1^{(n_1)}$ denote the composed tied map realized by iterating the reconstructed blocks according to the parsed schedule. By 0-BRH, we have exact functional equality

$$\boldsymbol{f}_\ell = \boldsymbol{B}_k^{(n_k)} \circ \cdots \circ \boldsymbol{B}_1^{(n_1)} = g_\ell,$$

i.e., the tied computation reproduces the full internal trajectory (not only the terminal output). Because $p$ encodes the same block parameters at the precision that defines $DLU$ and $U$ evaluates precisely the same composition, we obtain, for all inputs $x$,

$$U(p)(x) = g_\ell(x) = \boldsymbol{f}_\ell(x).$$

Finally, we get the runtime bound to derive the Levin Complexity. On the fixed machine $U$, a single evaluation of block $\boldsymbol{B}_j$ incurs time at most $c\, R(\boldsymbol{B}_j)$ for a machine dependent constant $c > 0$ (standard machine invariance up to constants; cf. the invariance discussion in Kolmogorov/Levin-style arguments (Kolmogorov, 1965; Levin, 1973)). Executing the schedule therefore costs

$$T(p) \le c \sum_{j=1}^{k} n_j\, R(\boldsymbol{B}_j).$$

Applying the runtime parity assumption $R(\boldsymbol{B}_j) \le (1+\delta)\, R(\boldsymbol{B_f})$ for all $j$ and using $R(\boldsymbol{f}_\ell) := \sum_{j=1}^{k} n_j R(\boldsymbol{B_f})$ gives

$$T(p) \le c\,(1+\delta) \sum_{j=1}^{k} n_j\, R(\boldsymbol{B_f}) = c\,(1+\delta)\, R(\boldsymbol{f}_\ell).$$

By the definition of Levin's time-bounded complexity

$$K_{\text{Levin}}(\boldsymbol{f}_\ell) \le |p| + \log T(p) + O(1),$$

and combining with the bounds from the previous steps yields

$$K_{\text{Levin}}(\boldsymbol{f}_\ell) \; \leq \; \sum_{j=1}^{k} DLU(\boldsymbol{\theta}(\boldsymbol{B}_j)) \; + \; O(k \log \ell) \; + \; \log R(\boldsymbol{f}_\ell) \; + \; \underbrace{\log c + \log(1+\delta) + O(1)}_{= \, O(1)}.$$

Absorbing the machine constant and the parity slack into $O(1)$ gives the stated inequality. □

## F  TRAINING LOSSSES AND ALGORITHM

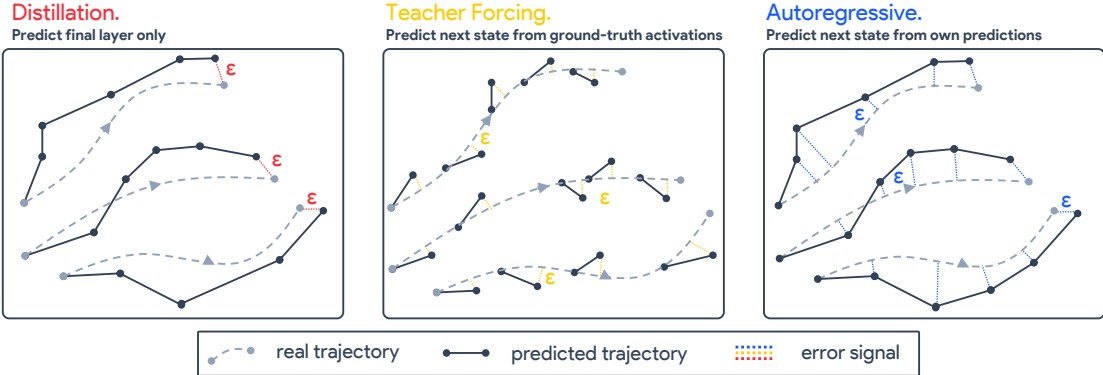

Figure 10: **Three training paradigms for learning recurrent approximations.** Each panel shows three token trajectories through depth. Gray dashed lines with filled circles represent the ground-truth teacher trajectories; black solid lines with filled circles show the student's predictions; colored dotted lines (with $\varepsilon$ labels) indicate the error signal between predicted and ground-truth states. **Left (Distillation):** The student network directly predicts the final layer from the initial state, with no supervision on intermediate representations. Error is measured only at the terminal state, providing no guidance on the representational trajectory. **Middle (Teacher Forcing):** At each depth step $\ell$, the student block predicts $\hat{\boldsymbol{x}}_{\ell+1} = \boldsymbol{B}(\boldsymbol{x}_\ell)$ using the ground-truth activation $\boldsymbol{x}_\ell$ from the teacher. Vertical arrows indicate where the student "resets" to ground-truth states. This enables efficient parallel training and prevents error accumulation, but creates a train-test mismatch since the model never learns to handle its own prediction errors. **Right (Autoregressive):** The student autoregressively predicts $\hat{\boldsymbol{x}}_{\ell+1} = \boldsymbol{B}(\hat{\boldsymbol{x}}_\ell)$ using its own previous predictions, matching inference conditions. Errors can compound across depth (shown by increasing deviation between trajectories), requiring the model to learn self-consistent, closed-loop dynamics. Our two-stage training (Sec. 3) combines both approaches: Stage 1 uses teacher forcing for stable, parallelizable pretraining; Stage 2 switches to autoregressive training to ensure self-consistency at inference.

```
a_gt = teacher(batch) # extract teacher activations
a_ar = a_gt[0] # start at first hidden state
loss = 0

for l in range(num_layers):
    b = get_block(l) # get recurrent block for phase
    out_tf, out_ar = b(a_gt[l]), b(a_ar) # dual forward passes

    # explicit hybrid loss
    l_tf = dist(out_tf, a_gt[l+1])
    l_ar = dist(out_ar, a_gt[l+1])
    loss += λ * l_tf + (1 - λ) * l_ar

    a_ar = out_ar # update student state

loss.backward()
optimizer.step()
```

Figure 11: **Hybrid training loop.** The model minimizes a convex combination of Teacher Forcing and Autoregressive errors, balanced by $\lambda$.

# G    SUPPLEMENTARY RESULTS

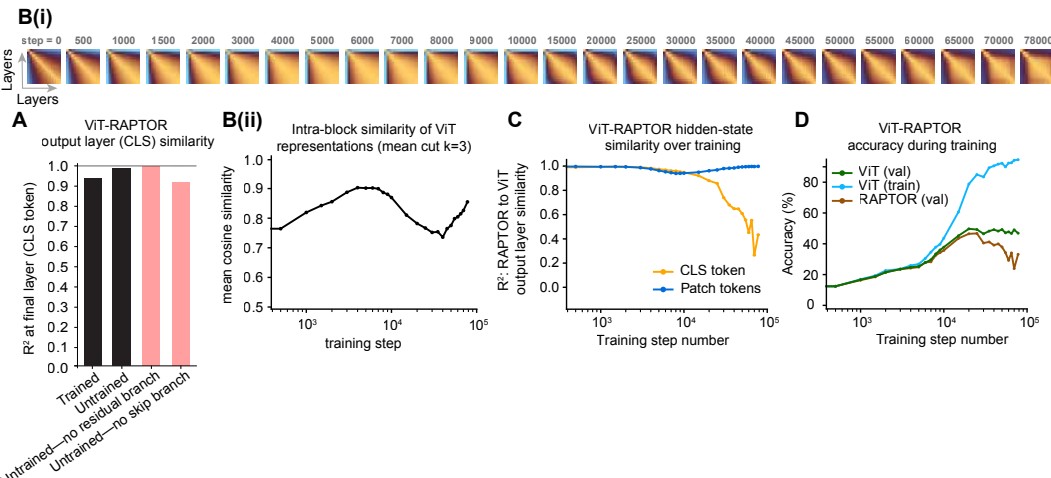

Figure 13: **Similarity and distillation dynamics during training and overtraining. A)** ViT-Raptor similarities between four pairs of ViT and Raptor networks. First from the left, a fully trained (but not overtrained) ViT network trained on CIFAR-100 and a Raptor student network trained (with k=3). Second, an untrained ViT network, and a Raptor student network. Third, an untrained ViT network with no residual branch in its architecture, so the entire network is just skip connections; and a Raptor student network. Fourth, an untrained ViT network with no skip connections, so the network no longer contains skip connections and is simply a feedforward network; and a Raptor student network. **Bi)** Layer-layer cosine similarities during training in a ViT network that is allowed to be overtrained (overtraining starts at around step 10000, see (D)). **Bii)** Mean intra-block cosine similarity with max-cut k=3 during training. **C)** Teacher-student Raptor reconstruction of ViT during training showing divergence after overtraining starts. **D)** CIFAR-100 accuracy in a ViT that was allowed to overtrain. Divergence between 'train' (green) and 'val' (blue) lines shows over training. X-axis is the same as in panels (B) and (c).

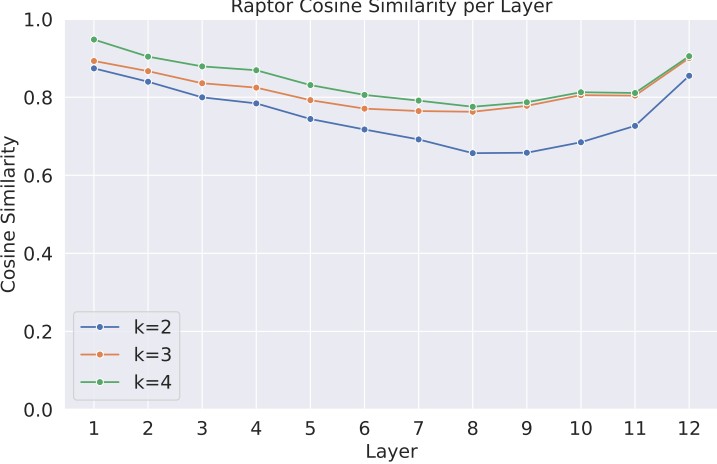

Figure 14: **Cosine similarity remains high through layers.** Cosine similarity between activations from Raptor models ($k = 2, k = 3, k = 4$) and activations from DINOv2-Base on the ImageNet-1k validation set. The consistently high similarity (mostly >0.7) indicates that Raptor effectively captures the dynamics of the original model, with the high alignment in the final layer confirming that the block-recurrent mechanism successfully approximates the target output representations.

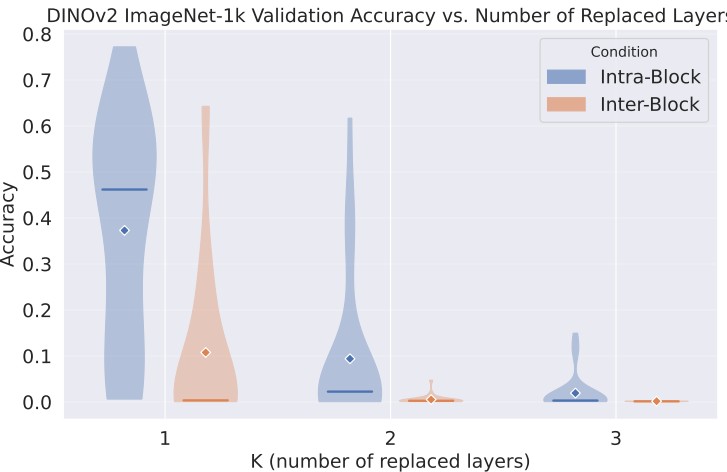

Figure 15: **Causal intervention.** DINOv2-Base accuracy on ImageNet-1k validation set with 1, 2, and 3 layers replaced with another layer ($k = 1, k = 2, k = 3$, respectively). Intra-block refers to replacing a layer with another layer from the same block. Inter-block refers to replacing a layer with a layer from a different block. Blocks are determined by the max cut algorithm. The significantly higher accuracy of intra-block replacements (blue) compared to inter-block (orange) confirms that layers within a block are functionally interchangeable in a way that any two arbitrary blocks are not, supporting the block-recurrent hypothesis.

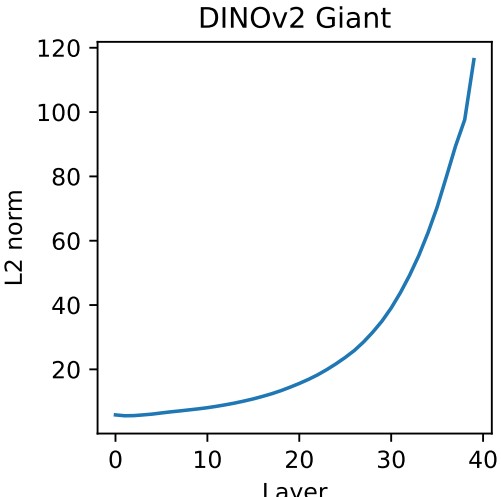

Figure 12: **Depth-wise feature norms.** Magnitudes grow with depth, motivating analysis on directions.

