# OpenReview forum: "Block Recurrent Dynamics in Vision Transformers"
_ICLR.cc/2026/Conference — ICLR 2026 Poster_

### Official Review · Reviewer_QZzj · 2025-10-30

**Soundness:** 3
**Presentation:** 3
**Contribution:** 3
**Rating:** 8
**Confidence:** 3

**Summary:**

This paper interprets vision transformers under a Block-Recurrent Hypothesis, investigating if the original computation that such models perform using L blocks can be achieved using k << L recurrent blocks while reproducing the internal activations of the full model. The paper then views these new models under a program of dynamical systems analysis.

**Strengths:**

I appreciate the very thorough account of the motivation, construction, and analysis of the Block-Recurrent Hypothesis. The mathematical definitions and propositions look sound, and the experiments around the identification of functional reuse and dynamical interpretability are well characterized. Understanding the dynamical structures within vision transformers that have so ubiquitously entered as de-facto models for visual tasks is interesting and necessary, and the paper does a good job at teasing some of that out. Specifically, the paper systematically explores the emergence of blocks using a controlled setting with a tiny vision transformer, and then scales the analysis to a larger DINOv2 ViT-base model. Additionally, it performs a series of experiments to understand dynamical properties of ViT tokens, offering a unique perspective on the appropriateness of directional (angular) geometry as a paradigm for dynamical analysis.

**Weaknesses:**

For table 1 and figure 5, it seems that the performance experiments were not repeated multiple times with different partitions or different random model seeds. I would have preferred to see some form of error bars quantifying how robust or varied the metric results are, and what to take from that. Also, in line 317, it says that accuracy saturates at k=4 without evidence to support that (what happens at k=5?).

In section 3, the manuscript states that the Raptor program will be tested on "Foundation models". What qualifies as a foundation model is highly debated between researchers, but regardless of the fact, simply testing on a ViT-base does not provide a convincing argument for the claim. For example, what happens when you scale to, say, a ViT-Large? Can Raptor be successfully applied to other objective functions outside of DINOv2, say MoCo-v3?

More generally, the authors show block dynamics to emerge across diverse ViTs. However, the ViTs that are tested seem to be trained on either a supervised loss or a contrastive loss. Do block dynamics emerge in a ViT that is trained on a local reconstruction loss, such as an MAE? How generalizable is the BRH?

**Questions:**

Most of the representational structure that is explored in this paper involves looking at the level of the output of transformer blocks. I would be curious to know how interactions within the attention matrices change across the model hierarchy, or whether you similarly observe functional reuse at the output of the attention layer in each block. Transformers can be thought of as possessing two computational pathways within each block, a fast one that is the result of the residual connection, and a slow one that is the result of a self-attention module. How do these two separate pathways interact with the Block-Recurrent Hypothesis, if at all? This is not a limitation of this current manuscript, but I would love to hear thoughts from the authors if they have any.

---

> ### Author Response · Authors · 2025-11-22
>
> We thank the reviewer for their thoughtful comments and the “Good Paper” rating. We address your specific concerns regarding robustness and scaling below.
>
> **Stability and Robustness**. The stability of Raptor is high and is quantified in Fig 1 and the new “fragmented” baseline experiments (added to Fig 3 in response to Reviewer qpb3). The updated Figure 3 can be seen here (where the fragmented blocks are referred to as `Random Shuffle’): [Updated Figure 3](https://raw.githubusercontent.com/anonymous123-user/raptor/refs/heads/main/max_cut_vs_random_accuracy.png)
>
> **Saturation at k=4**. The performance naturally saturates because at k=4, the blocks become small and approach the limit where the model simply reverts to a standard layer-wise ViT. As the precise shape of the convergence curve is not central to our work and we do not provide a quantification, we have removed this text.
>
> **Error Bars**. Finally, we agree with the reviewer that error bars on Table 1 and Figure 5 would be beneficial for completeness, although we expect them to be very small. We will include these results in the camera ready version of the paper.
>
> **Other Foundation Models.** We agree with the reviewer that the term “foundation model” is a broad term. We also agree that distillation of other foundation models like MoCo-v3 could strengthen our experimental suite and provide further support for the BRH. In this work, we focused on the DINOv2 ViT-B model because it is a standard benchmark for self-supervised learning with strong performance on numerous benchmarks.
>
> **MAE**. We agree that studying block dynamics in the context of ViTs trained with fundamentally different objective functions like MAE is an exciting avenue to explore to isolate if block recurrence is objective-dependent. More generally, we are also curious about expanding upon this work and identifying the causal mechanisms behind block recurrence. This work provided evidence that stochastic depth is a contributing factor (i.e. it strengthens block recurrence), which sets the foundation for future work to further investigate.
>
> **Q1**: This is an insightful question. The BRH makes a prediction about the final activation of each layer, but not specifically about the different computational streams in isolation (i.e. skip connections vs. residual connection attention + MLP). It would be interesting to investigate how these streams interact with respect to block recurrent dynamics. If one adopts the feature refinement view (see Greff & Schmidhuber 2016), it stands to reason that the attention and MLP are iteratively refining a feature whose identity is in the skip connection. The BRH models dynamical state as the output of each block, and our dynamical analysis (Fig 10) suggests that the update vector field collapses to low rank. This implies that fast and slow pathways in some way interact and cooperate to steer the token into the attractor basin.

---

### Official Review · Reviewer_oq2Q · 2025-11-01

**Soundness:** 3
**Presentation:** 4
**Contribution:** 3
**Rating:** 8
**Confidence:** 3

**Summary:**

The authors hypothesise (via their so-called Block-Recurrent Hypothesis) that ViTs operate in blocks, and that the computation of individual blocks may be compressible while allowing the same outputs.

They use the BRH's validation to train Raptor, which has weight-tied layers, almost acting like an RNN in the direction of transformers.

Finally, the authors offered a novel approach of 'dynamical interpretability' to analyse Raptor as a recurrent systems in the depth dimension

**Strengths:**

Really well motivated hypothesis, analysis and subsequent model design. Analysis was simple and straightforward - starting with representational similarity analysis then validating the connection to computational similarity by constructing Raptor. In turn, the construction/training objectives of Raptor were simple and easy to follow. The paper was a joy to read over all.

**Weaknesses:**

Introduction to different embeddings/tokens for those with a background in recurrent systems rather than ViTs would be very helpful.

**Questions:**

How many 'timesteps' was each recurrent block in Raptor run for? Table 3 provides layer splits, but could you clarify this more explicitly in the main text? For instance, does Block 1 in the k=3 configuration iterate 7 times to predict layers 1-7?

"Unlike classical distillation, which typically supervises logits (and occasionally a few intermediate “hints”), we enforce one-to-one alignment of all layers representations across the entire depth for the same inputs." - were distillation methods of this form compared against? Are there baseline compression methods that Raptor can be compared against?

The ε in the definition of BRH was never explicitly measured IIUC - does this vary across datasets/architectures?

---

> ### Author Response · Authors · 2025-11-22
>
> We thank the reviewer for the “Good Paper” rating and the helpful perspective from the recurrent systems community.
>
> 1. **ViT Explanation**. We particularly appreciate the suggestion to make the paper more accessible to those with a background in recurrent systems. We will include an explanation of ViTs, including a description of embeddings and tokens in the Appendix of the final version.
>
> 2. **Recurrent Timesteps**. The reviewer’s understanding is correct. To clarify, in the k=3 configuration, Block 1 runs for 7 timesteps (predicting layers 1-7), Block 2 runs for 3 timesteps (predicting layers 8-10), and Block 3 runs for 2 timesteps (predicting layers 11-12). We have made this explicit in the main text of our camera ready version.
>
> 3. **Distillation and Baselines**. We compared against the standard architectural compression (Small vs. Base). We did not compare against logit-distillation methods because our primary goal was functional equivalence (matching the trajectory) rather than just output matching. This approach provides a more direct validation of the BRH. However, for context, we do compare Raptor models against a standard DINOv2 ViT-S architecture, which has more parameters than our Raptor k=2 model and roughly 75% of the parameters of our Raptor k=3 model. We have added a note in our camera ready that while the focus of our work was on interpretability, the compression results against smaller ViTs are impressive. We have additionally included a parameter count comparison to Table 1.
>
> 4. **Measuring ε**. The reviewer is correct: ε is dependent on both the architecture and dataset. More precisely, ε is a theoretical bound that varies based on the norm of the activations; we included ε as an important part of the formalization of our hypothesis. Because raw ε values are difficult to contextualize across architecture and datasets (due to the aforementioned varying norm scales), we utilized downstream performance (e.g. ImageNet-1k classification accuracy) as a functional proxy. High performance indicates that ε is meaningfully low within the context of a given task.

---

### Official Review · Reviewer_89sj · 2025-11-01

**Soundness:** 4
**Presentation:** 3
**Contribution:** 3
**Rating:** 6
**Confidence:** 4

**Summary:**

This paper investigates the Block-Recurrent Hypothesis (BRH) in Vision Transformers (ViTs), which posits that the deep stacks of layers in a trained ViT can be decomposed into a small number of computational phases that repeatedly perform similar transformations. To operationalize this idea, the authors propose Raptor (Recurrent Approximations to Phase-structured Transformers), a framework that reconstructs a full ViT using only a few weight-tied recurrent blocks.

Instead of directly proving recurrence analytically, Raptor demonstrates the existence of block recurrence by example— training recurrent modules to reproduce not only the final outputs but the entire layer-wise activation trajectories of the teacher ViT. The resulting models achieve remarkable fidelity: with only k = 3 recurrent blocks, Raptor can recover ≈97% of DINOv2 ViT-B’s ImageNet-1k linear probe accuracy and a layer-wise R² of 0.73, indicating strong internal consistency between the reconstructed and original representations.

The work thus offers a constructive, empirical validation of the BRH, showing that the depth of ViTs can be “compressed” into a small number of iterative computational phases without major loss of functionality. It further opens the door to analyzing ViTs as dynamical systems, revealing phase-wise convergence behavior, token-specific dynamics, and low-rank collective motion in the representational flow across depth.

**Strengths:**

* Rather than relying on correlation or representational similarity alone, the authors operationalize the block-recurrence hypothesis by functionally reconstructing the ViT’s internal computations.

* Raptor replicates the performance and hidden representations of large ViTs (e.g., DINOv2 ViT-B) with only a few recurrent modules, across classification, segmentation, and depth estimation tasks.

* The two-stage Teacher Forcing + Autoregressive procedure, with gradual annealing and end-to-end refinement, shows careful engineering and clear causal analysis (e.g., TF-only collapse).

* The dynamical characterization—phase-wise convergence, angular attractors, weak contraction, and token-specific stability—offers valuable interpretability for transformer dynamics.

* The work suggests a compact, iterative view of transformer computation, resonating with ideas in dynamical systems, implicit networks, and recurrent neural architectures.

**Weaknesses:**

* The experiments mainly focus on image classification and linear probing; while segmentation and depth tasks are included, the demonstrations remain limited compared to the full spectrum of real-world applications.

* The use of max-cut–based phase segmentation is heuristic; the stability or uniqueness of the phase decomposition is not fully explored.

* The need for depth scaling indicates that the effective dynamics of ViT layers are non-stationary, so full recurrence may not hold uniformly across depth.

* While the recurrence hypothesis is supported empirically, it is not yet clear whether such recurrence can be directly exploited for efficiency or deployment in large-scale multimodal or generative models.

**Questions:**

The analysis and empirical validation of the block-recurrent hypothesis are quite compelling. However, has the team explored whether the recurrent compression (k ≪ L) can be leveraged for deployment efficiency—for instance, reducing parameters and FLOPs in large foundation models (e.g., MLLMs or multimodal ViTs) without sacrificing performance?

In principle, if a ViT can be represented with a few recurrent blocks, one might expect significant efficiency gains in inference-time or memory-limited environments. Has any investigation been conducted to measure these potential deployment benefits in practice?

Beyond classification, could Raptor-style recurrence be applied to dense or structured vision tasks (e.g., detection, captioning, video understanding)? It would be interesting to see whether the same “computational phase” structure holds when the model needs spatially adaptive or temporally extended reasoning.

Given the recent integration of ViT backbones into multimodal large language models, do the authors see any prospects for shared recurrent computation across modalities, or for cross-modal recurrence alignment?

Finally, could the dynamical interpretation of ViTs—especially the phase-wise attractor structure—help in regularizing or stabilizing the training of much deeper transformer-based architectures?

---

> ### Author Response · Authors · 2025-11-22
>
> We thank the reviewer for their detailed feedback. We address the concerns regarding scope and methodology below.
>
> - **Scope of Tasks**. We utilized standard vision foundation model benchmarks (ImageNet-1k classification, ADE20k semantic segmentation, NYUd-v2 depth estimation) to provide comparisons against a widely used baseline like DINOv2. We selected these experiments as necessary first steps in establishing and solidifying the BRH within a specific context before expanding to domains beyond pure vision.
>
> - **Max cut as a Heuristic**. Regarding any methodological concerns, we clarify that the Max-Cut algorithm gives an exact solution to phase segmentation via dynamic programming, or equivalently via brute-force search given that the number of layers is reasonably small (e.g., we distilled DiNOv2-base, which has 12 layers). While we utilized cosine similarity in this study, we agree that exploring other distance functions or methods for choosing blocks is a promising area to explore. At a broader level, we agree that, without theory proving the optimality of using Max-Cut on the representational cosine similarity matrix to define optimal block segments for recurrent distillation, this approach can fairly be considered a ‘heuristic’. However, we found that searching for optimal cuts within the representational similarity matrix proved to be an elegant approach that would otherwise require training large numbers of large neural networks. In addition, the relationship between these two spaces is central to the theoretical and experimental work we present. We show that these representational similarity blocks are far better than random grouping, and we conducted new experiments showing that these blocks are sensitive to permutations; these results empirically validate this heuristic and we are optimistic that future work will deepen our understanding of why representational blocks segment ideal recurrent computational blocks.
>
> - **Depth scaling**. When depth scaling is applied, we are explicitly modeling the network as a time-varying dynamical system rather than a strictly time-invariant system. This reformulation helps account for the non-stationarity of feature magnitude across layers (as shown in Fig 7. for the ViT-G model). However, we note that depth scaling is not strictly necessary for Raptor to function, as it only adds a modest boost to performance (see Table 2 for ablations). We hypothesize that this minor influence is due to the fact that a depth scaled Raptor is also only a minor deviation from purely time-invariant Raptor.
>
> - **Multimodal / Generative Models & Efficiency**. We agree that the recurrent structure of Raptor opens up exciting avenues to explore in different domains. Theoretically, the recurrent architecture has a reduced memory footprint which may be beneficial in edge applications where memory is limited.
>
> **Questions**:
>
> - **Recurrent compression** . While our primary contribution and interest is mechanistic interpretability, the efficient gains are real. Raptor with k=3 on ViT-B requires ~33% of the parameters while retaining ~97% accuracy. We also note that our final Raptor models converge in just 40 epochs on ImageNet-1k.
>
> - **Dense Tasks**: Raptor’s success on ADE20k (segmentation) and NYUd-v2 (depth) suggests that the “computational” phase structure holds for spatially adaptive tasks, and not just global classification.
>
> - **Multimodal models**: As discussed, we are also excited by the possible applications to other domains, and cross-modal recurrent alignment in particular. We thank the reviewer for highlighting this direction for future work.
>
> - **Regularization via Recurrent Weight Sharing**. We are particularly interested in this direction and are looking forward to leveraging recurrence as a useful inductive bias that might naturally regularize ViTs.

---

### Official Review · Reviewer_y5qx · 2025-11-03

**Soundness:** 3
**Presentation:** 4
**Contribution:** 3
**Rating:** 8
**Confidence:** 4

**Summary:**

This paper introduces the Block-Recurrent Hypothesis (BRH) for Vision Transformers (ViT): a pretrained ViT’s depth can be organized into contiguous “phases” such that the full computation can be reproduced by reusing a small set of shared blocks applied recurrently. The paper then instantiates the BRH by proposing Raptor, a weight-tied surrogate that uses max-cut and matches the activation of pretrained ViTs. Experiments on DINOv2 show that Raptor recovers most of the linear-probing accuracy of the original model with only a fraction of layers. Building on BRH, they further propose a dynamical interpretability view of depth, reporting directional (angular) attractors, token-specific dynamics, and low-rank update collapse in late layers.

**Strengths:**

1. From a hypothesis to an actionable algorithm along with interpretability insights, the paper is very insightful and can be very valuable to any follow-up research in ViTs.
2. Extensive experiments, both small and large scale, are presented to test the proposed hypothesis.
3. The paper is clearly written and well presented.
4. The stochastic depth section is very insightful and interesting.

**Weaknesses:**

1. The large-scale experiments only used DINOv2. While it is a very widely-used backbone, experiments on other models such as SigLIP could greatly strengthen the work and show its broader applicability.
2. ViTs are now extensively used in the context with vision-language models (VLMs) now. When tuned alongside the LLM, it is natural to think whether the proposed hypothesis still holds.
3. Compute tradeoffs are not fully quantified. While FLOPs per iteration is considered, wall-clock, memory behavior, and parallelism effects of recurrent blocks vs conventional ViTs aren’t thoroughly analyzed.

**Questions:**

See above.

---

> ### Author Response · Authors · 2025-11-21
>
> We thank the reviewer for the thoughtful comments and for their positive review of our paper.  We additionally appreciate the suggestions on how best to broaden the work. We provide a short response below directly addressing the reviewer’s individual questions.
>
> 1. **Other backbones (SigLIP).** We agree that SigLIP is an excellent candidate for next steps and downstream applications. We focused on DINOv2 (ViT-B) because it is a widely used backbone for self-supervised tasks, which suited our primary goal of testing our hypothesis (the BRH). While we cannot run a full SigLIP pre-training cycle within the time constraints of the rebuttal period, we note that the BRH relies on the depth-structure of the Transformer architecture itself, and does not make any specific assumptions on the objective function. To date, all of our empirical analysis suggests that the hypothesis applies equally across datasets, model sizes, and objectives (e.g. Fig. 1). As such, we would thus predict transferability to SigLIP.
>
> 2. **Raptor for VLMs / LLMs.** We agree that this is a promising direction. Given constraints on scope and time, we wish to constrain the scope of this work to the core analyses and mechanistic experiments, but share the reviewer’s enthusiasm in testing block-recurrent architectures in a variety of domains, VLMs in particular. As discussed above, the BRH characterizes ViT activations generally, and we predict transferability to related architectures.
>
> 3. **Quantifying Compute Tradeoffs.** While our primary focus with Raptor is studying ViTs from an interpretability perspective, rather than from a computational efficiency perspective, we acknowledge the importance of quantifying compute tradeoffs to contextualize the Raptor’s practical utility. Indeed, due to a reduced number of trainable parameters, Raptors maintain a slightly smaller memory footprint than their non-recurrent counterparts. Specifically, if a vision transformer has O(N\*L) parameters, where N is the number of parameters per transformer layer, and L is the number of layers, a RAPTOR only uses O(N\*K) parameters where K<<L is the number of blocks. During training, since the majority of memory usage is taken up by storing activations and gradients, this difference is typically not significant. However, in edge applications performing inference, this reduction may become valuable. We note that training memory usage may also be reduced with some minor modifications following the ideas of ‘Reversible Residual Networks’ (Gomez et al., 2017), later extended to ‘Reversible Transformers’ for language (Kitaev et al,. 2020) and vision (Mangalam et al., 2022). Specifically, rather than storing all activations required for backpropagation, if one uses an invertible transformer layer as the Raptor block, it would be possible to recompute activations during the backward pass, thereby enabling even lower memory footprint training applications. Finally, Raptor’s forward inference FLOPs are not significantly higher than a standard ViT-B backbone (e.g. DINOv2) since the block architecture is nearly identical and applied for the same number of iterations. For our Raptor models distilled from DINOv2, there is a small increase in FLOPs due to the use of the SwiGLU activation function, which uses a larger hidden layer in the MLP. In addition, we see a marginal increase in FLOPs when depth scaling is utilized; however, these computational requirements are typically dwarfed by the requirements of the remainder of the model. Moreover, during training, Raptor models converge in just 40 epochs on ImageNet-1k. In terms of parallelism, since Raptors replace the depth of ViT’s with recurrent operations, there is no change to the traditional processing pipeline – depth is an inherently sequential computation, as is recurrence. We have clarified these topics in the discussion section of the manuscript.

---

### Official Review · Reviewer_qpb3 · 2025-11-10

**Soundness:** 2
**Presentation:** 2
**Contribution:** 3
**Rating:** 4
**Confidence:** 3

**Summary:**

The paper introduces a fresh perspective on vision transformers, showing that their computational flow can be explained through a block-recurrent computation structure. To support their claims, the authors present several visualizations and training experiments. After establishing the connection between vision models and block-recurrent structure, the authors leverage this perspective to propose a new interpretability approach inspired by principles from dynamical systems, which can be used to better understand vision models.

**Strengths:**

- The paper presents a fresh view on the computational flow of vision transformers, providing a better understanding of these important models.

- As far as I know, the idea is novel and creative.

- The authors include extensive analyses, including ablation studies and interpretability experiments.

**Weaknesses:**

**W.1. Limited empirical analysis for the main claim:**

I believe the main idea in the paper (the recurrent block view) should be justified through additional experiments. In its current form, the argument relies heavily on the power of distillation, which is a very strong model compression technique. To verify that this is not the primary cause of the success of Raptor-like models, I suggest the authors conduct the following two experiments:

**(i) Block structure variation:** Train Raptor-like models with a similar number of blocks and layers per block, but where each block includes a non-contiguous (fragmented) set of layers (for example, instead of [1,2,3], [4,5], [6,7], use [1,4,6], [2,3,7], [5]). Please note that previous work [a] which use smart grouping found that sharing parameters between non-adjacent layers can be very effective, somewhat contradicting the results reported in this paper.

**(ii) Causal intervention analysis:** At test time (without fine-tuning or retraining), replace the weights of K layers with (a) weights from different layers within the same block, and (b) weights from random layers elsewhere in the model (not from the same block). If, for several K values, the accuracy under option (a) is consistently higher, this would provide much stronger support for the authors’ claim.


**W.2. Strengthen the connection to dynamical systems:**

From my perspective, the current motivation is somewhat vague. For example, a devil’s advocate might ask what differentiates the proposed approach from the well-known layer-by-layer analysis. I suggest that the authors include a dedicated section to motivate their decision to use dynamical systems. Incorporating technical terms and principles from the field of dynamical systems could make the argument more compelling and insightful.

**W.3. Minor:**

There are several visualization issues. For example, the colors in Fig. 6 (right) are unclear, with three similar shades of green. Additionally, the excessive use of vspace or other compression commands makes the text too dense.

[a] DYNAMIC LAYER TYING FOR PARAMETER-EFFICIENT TRANSFORMERS. David et al. ICLR 24

**Questions:**

Q.1. Do the authors think that their insights can shed more light on the connections between vision diffusion models and Raptor-like vision transformers, given that they used significantly fewer parameters and recurrent computation? Such insights could provide valuable information on the optimal trade-offs between FLOPs and parameters in vision models.

Q.2. What is the compute gap required to train Raptor models compared to the baseline? Please provide relative numbers to help illustrate how challenging it is to train such models.

---

> ### Author Response · Authors · 2025-11-21
>
> We thank the reviewer for the constructive feedback and for acknowledging the “fresh perspective” our work provides on ViTs. We recognize the reviewer’s concerns and address them by performing the suggested experiments and clarifying our theoretical framing in the text.
>
> **W.1 Validation of Mechanism vs Distillation.** We agree that distinguishing the architectural benefit of recurrence from the strength of distillation itself is critical. To isolate the effect of recurrence, we performed the following suggested experiments:
>
> - **(i) Block structure variation**: We trained Raptor-like models where blocks consisted of non-contiguous (fragmented) layers (e.g. layers [1, 4, 6] sharing weights) instead of contiguous layers. The contiguous Raptor model (specified using our Max-Cut method) significantly outperforms fragmented baselines, exceeding one standard deviation over 10 different random fragmentations. This new experiment demonstrates that the block-recurrence phenomenon we observe depends both on the unique parameters for each recurrent layer and intra-block contiguity. We have added these results to Figure 3 in our camera ready version, the updated version can be seen here (where the fragmented blocks are referred to as `Random Shuffle’): [Updated Figure 3](https://raw.githubusercontent.com/anonymous123-user/raptor/refs/heads/main/max_cut_vs_random_accuracy.png)
> - **(ii) Causal intervention analysis**: We conducted weight-swapping experiments on a pretrained DINOv2 (ViT-B) network. Replacing layer weights with weights from the same block (option (a)) yields significantly higher accuracy on ImageNet-1k than replacing weights with those from different blocks (option (b)). This confirms that layers within a block are functionally interchangeable in a way that layers across blocks are not. These results isolate block-recurrence as the mechanism driving the observed performance. These new results are included in the Appendix of our camera ready version, and can be seen in the new figure here: [Causal Intervention Analysis](https://raw.githubusercontent.com/anonymous123-user/raptor/refs/heads/main/causal_intervention.png)
>
> We believe that the results of these new experiments are fully in agreement with our other findings and reinforce our hypothesis that block-recurrence is an emergent phenomenon within these networks and cannot be explained by general distillation. We agree that these results are on the surface different from those reported in prior work [a]; however we note that that work leverages a dynamic reinforcement learning trained layer-tying mechanism which is significantly more complex than our static max-cut method. It is possible that future work could leverage a sophisticated RL agent in a similar manner to dynamically allocate recurrent Raptor blocks and see a significant benefit. We thank the reviewer for identifying and suggesting these experiments.
>
> **W.2. Connection to Dynamical Systems.** The reviewer asks us to clarify how our analysis of the representational dynamics observed across layers differs from ‘well-known layer-by-layer analysis’. We acknowledge that our original text did not develop an explicit definition of across layer dynamics; we will address this with additional text clarifying our terms. Briefly, the approaches used in the latter half of the paper analyze the rules of evolution (our operational definition of ‘dynamics’) that transform representations from layer to layer. By identifying attractor-dynamic-like behavior, we move beyond describing static snapshots or pairwise comparisons of layer representations to describing the mechanism of representational convergence. We have added a paragraph to Section 4 in the manuscript that contrasts “feature descriptive” vs. “dynamical” analysis.
>
> **W.3. Minor Visualization issues.** We appreciate the reviewer pointing out these issues. We have adjusted the color palette in Fig. 6 and fixed the spacing in the final manuscript.

---

> > ### Author Response · Authors · 2025-11-21
> >
> > **Q1**: Indeed, our experiments suggest that ViTs iteratively refine their representations via recurrent application of the same operator within a block, which is similar to diffusion denoising. Furthermore, given the adoption of vision transformer architectures for diffusion (e.g. the Diffusion Transformers of Peebles & Xie (2023)) our analysis and method likely apply in this scenario as well.
> >
> > **Q2**: Raptor is highly efficient to train, requiring only ~40 epochs to distill DINOv2 (ViT-B) on ImageNet-1k. Specifically, each Raptor model trains for 625k gradient steps with a batch size of 64 on ImageNet-1k, whereas DINOv2 distillation requires 625k steps with a batch size of 2048 on LVD-142M. Direct comparison is challenging due to differences in dataset scale and the distillation hierarchy: DINOv2 distills base-from-giant, while Raptor distills recurrent-base-from-base. Nevertheless, these numbers provide a useful reference point. Specifically, Raptor’s ability to achieve high performance with an equal step count, despite a significantly smaller batch size and dataset, highlights the promising potential of this approach.

---

> > > ### Comment · Reviewer_qpb3 · 2025-11-22
> > >
> > > I thank the authors for addressing most of the concerns, especially Weaknesses 1 and 2.
> > >
> > > The new experiments isolate the impact of distillation and further strengthen the claims.
> > >
> > > I am happy to increase my score.

---

### Meta-Review · Area_Chair_iJTG · 2025-12-30

**Summary:**

The main reviewer consterns are aroundempirical breadth, clarity of the dynamical framing, robustness, and scope. Reviewer qpb3 initally felt that the “recurrent block view” was not sufficiently justified, worrying that the results might be due mainly to the strength of distillation, and asked for block‑structure variation and causal intervention tests; they also requested a clearer distinction from standard layer‑by‑layer analysis and noted some visualization issues. Reviewer y5qx notes that large‑scale experiments “only used DINOv2,” suggests that additional backbones like SigLIP and VLM settings would strengthen generality, and points out that compute tradeoffs (wall‑clock, memory, parallelism) are not fully quantified. Reviewer 89sj mentions that segmentation and depth experiments are limited, that max‑cut phase segmentation is heuristic and its uniqueness/stability is underexplored, that depth scaling hints at non‑stationary dynamics, and that deployment benefits in multimodal or generative settings are not yet shown. Reviewer oq2Q’s only concrete weakness is that some readers from recurrent‑systems backgrounds might need a clearer introduction to ViT embeddings and tokens.

The AC recommends acceptance, following reviewer majority opinion, because the paper offers a well‑formulated hypothesis, a constructive instantiation (Raptor) with strong empirical support, and a coherent dynamical analysis that deepens our understanding of ViT depth computation. The additional experiments disentangling recurrence from generic distillation, the causal weight‑swap analysis, and the clarified discussion of dynamics and compute strengthen the work substantially. While breadth across architectures and tasks could be expanded in future work, the present results on DINOv2 and small ViTs already provide a compelling case for block recurrence and open a promising avenue for mechanistic analysis of transformers. On balance, AC agrees with positive points raised by all reviewers which outweigh the negative points. The authors are strongly encouraged to include the additional reviewer recommendations, experiments from rebuttal and clarifications in the camera-ready version.

**Reviewer Concerns:**

The rebuttal directly addresses the critical points. For Reviewer qpb3, the authors run the requested experiments: they train Raptor‑like models with non‑contiguous “fragmented” blocks and show that max‑cut contiguous blocks significantly outperform random shuffles, and they perform causal interventions on DINOv2 by swapping weights within versus across blocks, finding clearly higher accuracy when swaps stay within the same block. Reviewer qpb3 explicitly states that these new experiments “further strengthen the claims” and that they are “happy to increase my score.” The authors also clarify what they mean by “dynamics” versus descriptive layer‑wise analysis and fix visualization issues. For Reviewer y5qx and Reviewer 89sj, they provide more discussion of compute: Raptor uses O(N·k) parameters versus O(N·L), converges in ~40 epochs on ImageNet‑1k, and has similar FLOPs to the teacher, with recurrence not changing the sequential nature of depth; they clarify that their focus is interpretability rather than efficiency, while acknowledging future applications and noting that current dense tasks (ADE20k, NYUv2) already show the phase structure extending beyond classification. For Reviewer QZzj, they clarify the number of timesteps per block and remove an unsupported saturation claim, and they agree that more backbones and objectives (e.g., MAE, MoCo‑v3, VLMs) are natural next steps rather than part of this first study. The question of full generalty to other training objectives and multimodal models and the optimality of max‑cut segmentation remains open, but is clearly framed as future work rather than over‑claimed.

**Reviewer Scores:**

Given the rebuttal and follow‑up comments, it is reasonable to expect that Reviewer qpb3 has raised their rating above 4, as they explicitly mention increasing their score. Reviewers y5qx, 89sj, and oq2Q already recommend acceptance (8, 6, 8) and express only moderate reservations about scope and generalization, which are natural for a mechanistic study on a major backbone. The remaining concerns—limited backbones (only DINOv2 at scale), heuristic phase segmentation, and lack of immediate deployment results—do not undermine the central contribution of empirically validating the Block‑Recurrent Hypothesis and demonstrating that a small number of recurrent blocks can faithfully reconstruct a deep ViT’s internal dynamics and linear‑probe performance. The interpretability program based on angular geometry, attractor‑like behavior, and low‑rank update collapse is also a clear asset and aligns well with the ICLR interpretability track.

---

### Decision · Program_Chairs · 2026-01-26

Accept (Poster)